# Brain sex-dependent alterations after prolonged high fat diet exposure in mice

Valentina Murtaj [1,2,9,13], Silvia Penati[3,4,10,13], Sara Belloli[2,5], Maria Foti[6], Angela Coliva[2], Angela Papagna[6], Cecilia Gotti[7], Elisa Toninelli[2], Remy Chiaffarelli [2,6,11], Stefano Mantero [8,12], Susanna Pucci[7], Michela Matteoli [3,4], Maria Luisa Malosio [3,4✉] & Rosa Maria Moresco [2,5,6✉]

We examined effects of exposing female and male mice for 33 weeks to 45% or 60% high fat diet (HFD). Males fed with either diet were more vulnerable than females, displaying higher and faster increase in body weight and more elevated cholesterol and liver enzymes levels. Higher glucose metabolism was revealed by PET in the olfactory bulbs of both sexes. However, males also displayed altered anterior cortex and cerebellum metabolism, accompanied by a more prominent brain inflammation relative to females. Although both sexes displayed reduced transcripts of neuronal and synaptic genes in anterior cortex, only males had decreased protein levels of AMPA and NMDA receptors. Oppositely, to anterior cortex, cerebellum of HFD-exposed mice displayed hypometabolism and transcriptional up-regulation of neuronal and synaptic genes. These results indicate that male brain is more susceptible to metabolic changes induced by HFD and that the anterior cortex versus cerebellum display inverse susceptibility to HFD.

[1] PhD Program in Neuroscience, University of Milano-Bicocca, Monza (MB), Italy. [2] Department of Nuclear Medicine, IRCCS San Raffaele Scientific Institute, 20132 Milan, Italy. [3] Institute of Neuroscience, National Research Council of Italy (CNR) c/o Humanitas Mirasole S.p.A, Via Manzoni 56, 20089 Rozzano (MI), Italy. [4] Laboratory of Pharmacology and Brain Pathology, Neuro Center, IRCCS Humanitas Research Hospital, Via Manzoni 56, 20089 Rozzano (MI), Italy. [5] Institute of Molecular Bioimaging and Physiology, CNR, 20090 Segrate (MI), Italy. [6] Department of Medicine and Surgery, University of Milano-Bicocca, 20900 Monza (MB), Italy. [7] Institute of Neuroscience, National Research Council of Italy (CNR) c/o Università di Milano-Bicocca, Via R. Follereau 3, 20854 Vedano al Lambro (MB), Italy. [8] Institute for Genetic and Biomedical Research, National Research Council of Italy (CNR) c/o Humanitas Mirasole S.p.A, Via Manzoni 56, 20089 Rozzano (MI), Italy. [9] Present address: Neuroimmunology Unit, Institute of Experimental Neurology, IRCCS San Raffaele Hospital and Vita Salute San Raffaele University, Milan, Italy, 20132 Milan, Italy. [10] Present address: Department of Pathology and Immunology, Washington Univerisity School of Medicine, St. Louis, MO 63110, USA. [11] Present address: Werner Siemens Imaging Center, Department of Preclinical Imaging and Radiopharmacy, Eberhard Karls Universität Tübingen, 72076 Tübingen, Germany. [12] Present address: DCSR, National Research Council of Italy (CNR), Via A. Corti 12, 20133 Milan, Italy. [13] These authors contributed equally: Valentina Murtaj, Silvia Penati. ✉email: marialuisa.malosio@in.cnr.it; rosa.moresco@unimib.it

Obesity represents a critical health and social problem worldwide linked to increased risk of developing cardiovascular diseases, type 2 diabetes, cancer, and brain disorders. Weight gain is associated with the so-called metabolic syndrome (MetS) characterized by insulin resistance (IR), mild-to-frank hyperglycemia, hyperinsulinemia, modification of immune cell metabolism, and low-grade systemic inflammation[1]. Liver is particularly vulnerable to excessive caloric intake, with lipid intermediates determining metabolic modifications and favoring IR and hepatic disorders[2].

Sex difference in susceptibility to metabolic diseases is a still poorly investigated field along with the understanding of sex specific brain responses to pathological conditions. Despite a similar prevalence of MetS, males and females differ in the risk factors determining this condition[3]. Clinical studies showed that obese females are at lower risk of developing non-alcoholic fatty liver disease (NAFLD) than males, but once established, have a higher probability of advanced liver fibrosis, particularly after the age of 50 years[4]. The age-specific prevalence of late-life dementia is higher in women than in men[5]. Midlife women with MetS showed a larger decline of perceptual speed over a decade compared to normal ageing[6].

An aberrant feeding behavior, along with increased intake of specific nutrients, leading to obesity, overweight, and IR may be a concurrent factor for altered regional brain functions, cognition and behavior[7]. The mechanism is not fully understood but several findings suggest that hyperglycemia and IR influence brain cell metabolism and mitochondrial activity, favoring an overall pro-inflammatory environment that increases oxidative stress[8,9]. In fact, in addition to the peripheral alterations, leading to systemic inflammation[10], high-fat diet (HFD) induces a central reactive astrogliosis that involves microglial activation and astrocytosis at early stages[11]. Increased number of microglial cells has been reported in the hypothalamus of HFD mice and in obese subjects, which has been associated with a dysregulation of satiety mechanisms[12,13]. Interestingly, some evidence suggests that increased neuro-inflammatory responses in this region are sex dependent and driven by metabolic modifications related to cholesterol and sex hormones[14,15].

To better understand how obesity and impaired glucose tolerance affect the brain in a sex dependent manner, the effects of a prolonged exposure to HFD on cerebral glucose metabolism, inflammation and synaptic proteins were evaluated using in vivo positron emission tomography (PET), western blotting and RNA expression analyses in the anterior cortex and cerebellum. Anterior cortex was selected for molecular analysis, because it is involved in cognitive function related to food preferences, eating disorders and anxiety[16]. Cerebellum was added to transcriptomic analysis on the basis of the PET metabolic results and of volumetric modification reported from imaging studies on obesity[17]. PET findings from different cerebral areas were correlated with blood hemato-chemical profiles in HFD compared to normal fed mice. For the study we selected two different HFD regimens (45% and 60%) and used the C57Bl/6J strain prone to develop obesity and diet induced-IR[18].

Our results indicate that male brain is more susceptible to metabolic changes induced by HFD and that the anterior cortex versus cerebellum display inverse susceptibility to HFD.

## Results
**HFD reduces glucose tolerance and progressively leads to metabolic modifications resembling metabolic syndrome.** Animals fed with different HFD (45% and 60%) showed a progressive increase in body weight, which was higher and faster in males compared to females (Fig. 1a, b and Supplementary

Data 1). The BMI of HFD treated male mice was already significantly higher compared to controls at 7 and 12 weeks of diets (Supplementary Fig. 1 and Supplementary Data 1), however the BMI at 33 weeks of diet was significantly increased in both sexes in the 60HFD compared to STD control group (Fig. 1c and Supplementary Data 1). At 31 weeks, the glucose tolerance test (GTT) curve of HFD male (Fig. 1f top and Supplementary Data 1) and female (Fig. 1f bottom and Supplementary Data 1) mice was higher compared to STD at all time points (Fig. 1f). Area under the curve (AUC) of glucose concentrations confirmed a significant difference in both sexes and diets compared to control groups (Fig. 1g and Supplementary Data 1).

In both male and female mice, a significant increase in serum cholesterol appeared already at 14 weeks of HFD compared to controls; however, males reached significantly higher concentrations than females: 45HFD: $p$ value <0.0001 F vs M 14 weeks; $p$ value <0.0001 F vs M 33 weeks; 60HFD: $p$ value <0.0001 F vs M 14 weeks; $p$ value <0.0001 F vs M 33 weeks (Fig. 1d, e and Supplementary Data 1). HDL and LDL sex differences were found in the 60HFD group only, with males showing significantly higher serum concentrations compared to females (Supplementary Fig. 2a, b and Supplementary Data 1). In males but not in females, aspartate aminotransferase (AST) and alanine aminotransferase (ALT) liver enzymes were significantly increased at 33 weeks. The difference of AST and ALT between males and females on HFD was significant (Supplementary Fig. 2c, d and Supplementary Data 1). No modifications induced by HFDs were detectable for albumin (Supplementary Fig. 2e and Supplementary Data 1). The liver sections of both sexes fed with both HFDs showed diffuse lipid droplets indicating steatosis (Fig. 1h). Based on these data we conclude that HFD mice have a phenotype resembling MetS.

**HFD induces sex-dependent regional modifications in brain glucose uptake associated with MetS status.** At 31 weeks, the global uptake of [18F]-FDG was similar in both sexes fed on STD and HFD diets (Fig. 2a and Supplementary Data 1). Therefore, as performed in clinical research, mean values of the whole brain [18F]-FDG uptake were used to normalize regional data. A significant increment of normalized [18F]-FDG uptake occurred in olfactory bulbs of males and females on 60HFD (Fig. 2b and Supplementary Data 1); in addition, males showed alterations also in the anterior cortex and cerebellum (Fig. 2c, d and Supplementary Data 1).

Spearman's correlations between brain metabolism and BMI or metabolic markers revealed sex and regional differences. In males (Fig. 2e and Supplementary Data 1), normalized [18F]-FDG uptake of brain regions showing metabolic alterations were significantly correlated, either positively or negatively, with BMI and hemato-chemical parameters. Cholesterol and BMI positively correlated with olfactory bulbs and negatively with cerebellum glucose metabolism. Finally, [18F]-FDG uptake in the whole brain was negatively correlated with glycemia, LDL, ALT and AST. In females (Fig. 2f and Supplementary Data 1), ALT was positively correlated with olfactory bulb and negatively with cerebellum metabolism, while striatum was positively associated with AST. Coefficients $r$ and $p$ values for all correlations are reported in Supplementary Table 3. Overall, in males, the brain regions showing glucose metabolism alterations are those displaying significant correlations with peripheral metabolic parameters. In females only liver enzymes (AST, ALT) correlated with [18F]-FDG uptake in the olfactory bulb, striatum and cerebellum.

**HFD promotes a brain inflammatory phenotype that is differently influenced by sex and metabolic state in the periphery.** Longitudinal analysis revealed, in male fed with 45HFD or

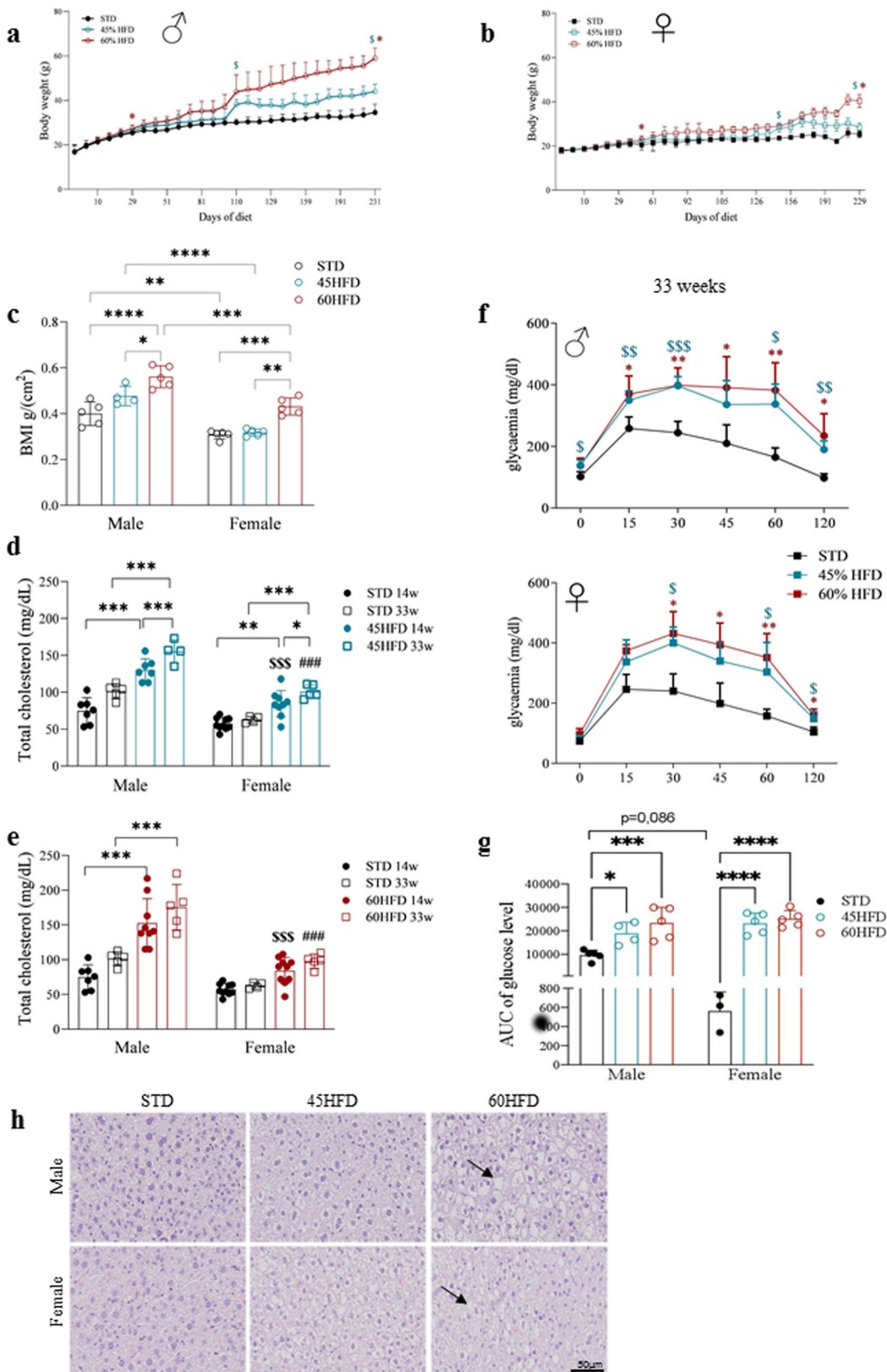

60HFD, a time-dependent increase in the uptake of radioligand [¹⁸F]-VC701, detecting TSPO-related inflammation (Fig. 3a–c left panels, Fig. 4a, and Supplementary Data 1). In females, this effect was present only for 60HFD (Fig. 3c right panel and Fig. 4a). These results were partially confirmed in postmortem analysis of anterior cortex. A significant increase in Iba-1 level, a marker of microglia/macrophages, was detected in 60HFD male mice and a trend toward increase in 45HFD compared to STD. In females, a significant increase was present only in the 45HFD group (Fig. 4b and Supplementary Data 1). In males, [¹⁸F]-VC701 uptake exhibited a positive correlation with total and LDL cholesterol and ALT in different brain regions including anterior cortex,

**Fig. 1 High-fat diet mouse model characterization and sex differences.** Body weight (**a** males; **b** females), *$p \leq 0.05$ 60% HFD vs STD, $$p \leq 0.05$ 45% HFD vs STD (statistical significance in **a**, **b** was maintained from the indicated time points until sacrifice). **c** BMI in males and females at 33 weeks of diet. **d** Total cholesterol serum levels in males and females 45HFD vs STD controls at 14 and 33 weeks of diet. **e** Total cholesterol serum levels in males and females 60HFD vs STD controls at 14 weeks and 33 weeks of diet; STD controls are the same for 45/60HFD vs STD males analysis and for 45/60HFD vs STD females analysis; for both **d**, **e** $^{$$$}p \leq 0.001$ 14-weeks HFD male vs female; $^{###}p \leq 0.001$ 33-weeks HFD male vs female; **f** glucose tolerance test in male and female mice at 31 weeks of diet; *60HFD vs STD, $45HFD vs STD. **g** Total area under the curve (AUC) of glucose level at 33 weeks of diet in male and female mice. **h** Hematoxylin and eosin staining of liver sections in STD, 45% HFD, and 60% HFD in males (upper panels) and females (lower panels) at 14 weeks of diet, black arrow indicates lipid droplets. Data are expressed as mean ± SD. Data in **a**, **b**, **f** were analyzed using one-way ANOVA for repeated measures with diets and time points as variables. Data in **c**, **g** were analyzed using two-way ANOVA including sex and diet as variable. Data in **d**, **e** were analyzed using three-way ANOVA, in which diet, sex, and time point were considered. Multiple statistical tests were followed by Tukey's post hoc test for multiple comparison; */$$p \leq 0.05$, **/$$p \leq 0.01$, ***/$$$/###$p \leq 0.001$, ****$p \leq 0.0001$.

hippocampus, hypothalamus, and cerebellum (Fig. 4c and Supplementary Data 1). No correlations were detected in females (Fig. 4d, Spearman's r and p values, Supplementary Table 4, and Supplementary data 1). GFAP protein levels investigated by western blots in ACX did not reveal any significant difference among groups. (Supplementary Fig. 5 and Supplementary Data 1). Altogether these results show that males are more prone to accumulate [18F]-VC701 at 31 weeks with all types of diets and this correlates with peripheral metabolic parameters. In females despite an increase in [18F]-VC701 uptake at 31 weeks in 60HFD no correlations with metabolic parameters were observed.

**Differential gene expression in the anterior cortex and cerebellum of mice shows both common and distinct effects of HFD in males and females.** Principal component analysis (PCA) of female and male transcript showed a clear separation between cortex and cerebellum (Supplementary Fig. 8). Long-term exposure to either HFD regimen produced gene transcript profile alterations in both sexes in the anterior cortex (ACX, Fig. 5a–f). Gene Ontology (GO) and Kyoto Encyclopedia of Genes and Genomes (KEGG) Pathway analyses on 1215 60HFD differentially expressed genes (DEG) in males and 2360 in females showed an enrichment for functional annotation categories (GO) including, common to both sexes: *nervous system development, positive regulation of synaptic transmission, modulation of chemical synaptic transmission* (Supplementary Fig. 3a, c); *specific to males: positive regulation of excitatory postsynaptic potential, cytoskeleton organization, transmembrane receptor protein tyrosine kinase signaling pathway* (Supplementary Fig. 3a); *specific to females: positive regulation of neuron projection development, vesicle mediated transport* (Supplementary Fig. 4a). Common KEGG pathway for males and females are *synaptic vesicle cycle, glutamatergic synapses, protein processing in endoplasmic reticulum, GABAergic synapse* (Supplementary Fig. 3b, d). GO and KEGG enrichment pathways for males and females on 45HFD are reported in Supplementary Figs. 3c, d and 4c, d, respectively. In addition, application of Ingenuity Pathway Core Analysis (IPA, Qiagen) revealed neuro-specific significantly altered pathways ($-\log (p$ value) $\geq 1.3$) in both males and females exposed to 60HFD, including *Synaptogenesis Signaling Pathway, GABA Receptor Signaling, Synaptic Long-Term Depression* and *Long-Term Potentiation, CREB signaling in neurons, Glutamate Receptor signaling, Axonal guidance, Glutamate degradation* and *Dopamine receptor signaling* (Supplementary Fig. 6 and Supplementary Data 1). The presence of pathways involved in *Insulin secretion, Leptin, glycolysis, Glycogen degradation, MODY* and *AMPK*, confirmed the derangement of several metabolic functions also in the anterior cortex of HFD fed mice. Finally, also pathways related to basic cellular mechanisms, such as *Unfolded Protein Response* (UPR), *Clathrin-mediated endocytosis, tight junction, actin cytoskeleton, RhoGDI, Senescence, PTEN, mTOR,* and *GADD45 signaling* emerged as differentially regulated in

HFD animals. All of the identified pathways, except for PTEN and UPR signaling, showed a negative z-score indicating a reduction of transcript expression (Supplementary Fig. 6 and Supplementary Data 1).

Upon manual annotation, 95 neuronal genes turned out as significantly altered in males fed on 60HFD, whereas 121 were detected in females (Fig. 6a and Supplementary Data 1). Functional categorization of these genes identified 19 categories based on protein functions (Fig. 6b, c and Supplementary Data 1). In both males and females only few neuro-related genes were up-regulated (4 in males, 10 in females). In both males and females, transcripts associated with synaptic functions and ion channels were down regulated showing common and sex specific modifications (Fig. 6a and Supplementary Data 1). It is interesting to note that synaptic organizers cerebellin1 (−6.61-fold) and erebellin3 (−2.7-fold) transcripts were specifically down-regulated only in males. These data altogether suggest that in the ACX of both males and females fed with 60HFD a reduction of mRNAs encoding proteins involved both in basal and stimulated neuronal excitability occurs.

Beside neural genes, several insulin-like growth factor binding proteins (2, 5, 7 in males and 2, 5, 6, 7 in females) were down regulated in both sexes whereas protein translation elongation and initiation factors (Eef1a2, Eif2b1,2,3, Eif3c, Eif4a3, Eif4h) were down-regulated only in males. Furthermore, transcripts coding for protein palmitoylating enzymes (Zdhhc13, 16, 20 and 21), along with genes involved in mRNA processing (Ago3, Rbm3&4, Lin28B, Lsm4, Dcp2, Hfm1, DDX31, Papd4, PolR2K) and inflammosome (Nlrp 5 in female and Nlrp9b in males) were up-regulated in both sexes. Concerning markers of BBB integrity, Adora2a transcript encoding the adenosine A2 receptor ($A_{2A}R$) was up-regulated in males along with a reduction in claudin 20 levels, while in females an important down-regulation of claudin 5 (−12.41) and claudin 11 (−5.03) transcripts was observed. The microglia chemokine CX3CL1/Fractalkine and its receptor CX3R1 were significantly down-regulated in females but not in males. Another important class of down-regulated transcripts was the low-density lipoprotein receptor-related proteins (Lrp), in particular Lrp1 that was significantly reduced in both sexes, whereas Lrp3, Lrp6 and Lrp10 were reduced only in females and Lrp4 only in males. Considering genes involved in metabolism, glycogen phosphorylase B (Pygb) was down-regulated in both sexes (−3.98-fold in males and −2.31-fold in females) whereas only females show an important reduction of glycogen phosphorylase M (Pygm, −15.43-fold) transcript. Interestingly, in females but not in males two pyruvate dehydrogenase kinase isoenzymes (Pdk2&3) were down-regulated (−4.13, −2.22). On the contrary, higher levels of pyruvate dehydrogenase phosphatase regulatory subunit, Pyruvate dehydrogenase kinase 1 (Pdk1, +2.85) and Lactate dehydrogenase-b (Ldhb, +2.54) were lower in females but not males. In males an up-regulation (+2.35) of the monocarboxylic acid transporter Slc16a5 mRNA was observed.

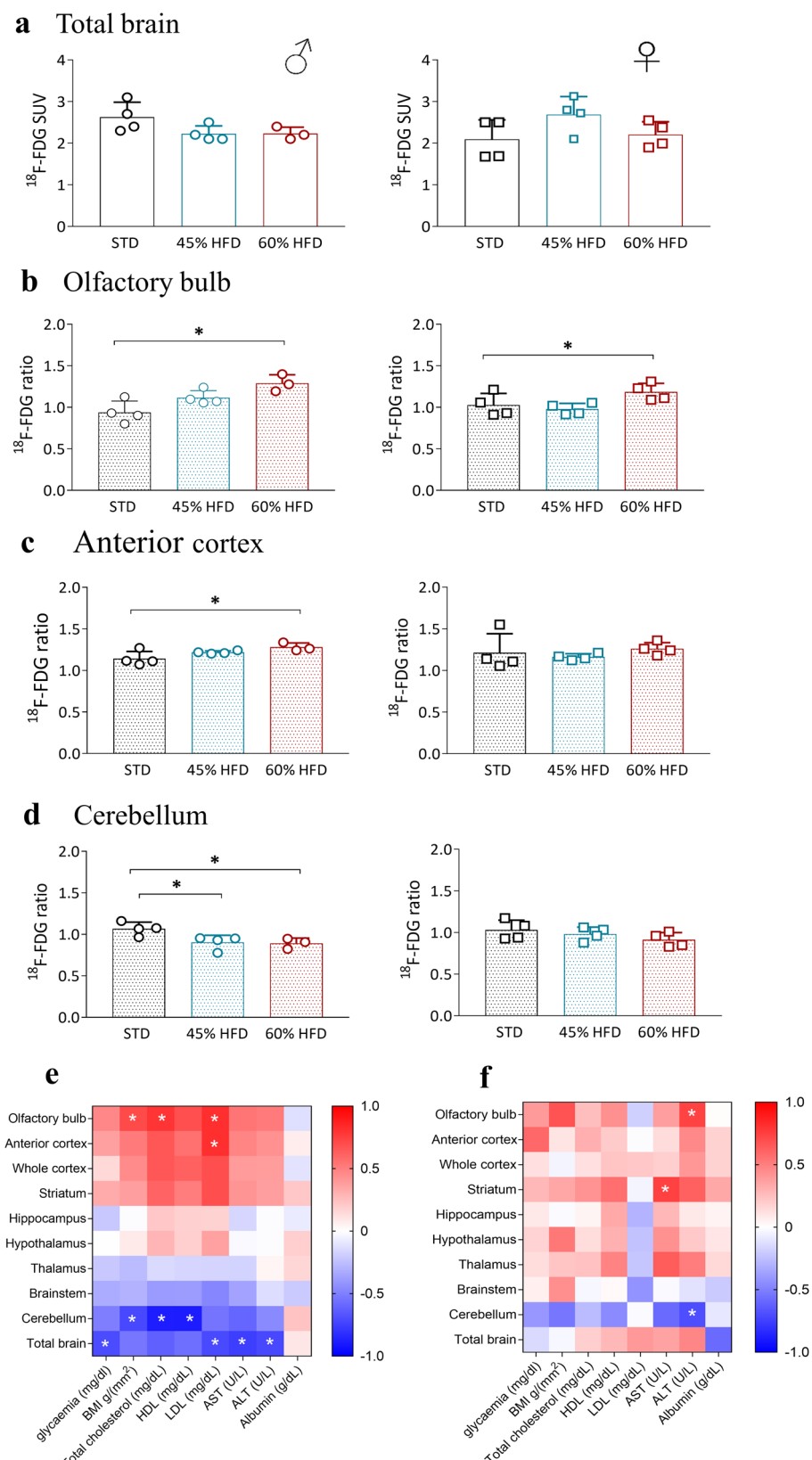

Finally, in males 16 olfactory receptors genes were up-regulated by HFD, whereas in females only two were altered in the opposite direction.

Cerebellum (Cb) transcript analysis was performed given the opposite pattern of FDG-uptake with respect to ACX. Also in the Cb long-term exposure to either HFD regimen produced

remarkable effects on gene transcript profiles in both sexes (Fig. 5g–l). In males on HFD60 compared to STD 1039 DEGs were found (935 down- and 104 up-regulated), among the down-regulated were 455 predicted genes, 91 vomeronasal-1 and 2 receptors and 168 olfactory receptors. Similarly, in Cb of males fed on HFD45 1738 DGEs (1251 down- and 487 up-regulated)

**Fig. 2 PET imaging evaluation of glucose metabolism in HFD mice. a** [18F]-FDG total brain glucose uptake, standardized uptake value (SUV), in males (left) and females (right) at 31 weeks; data are expressed as mean ± SD of SUV radiotracer uptake. [18F]-FDG glucose ratio to total brain uptake in males (left) and females (right) olfactory bulb (**b**), anterior cortex (**c**), and cerebellum (**d**) at 31 weeks. Data are expressed as mean ± SD of ratio over total brain radiotracer uptake; All data were analyzed using one-way ANOVA followed by Dunnett's post hoc test for multiple comparison (*$p \leq 0.05$). **e, f** Heat map of [18F]-FDG brain uptake Spearman's $r$ correlations with basal glycemia, BMI, and serum parameters at 31 weeks of diet in males (**e**, left) and females (**f**, right) mice. The pseudocolor bar represents the values ranging from −1 (blue, negative correlation) to +1 (red, positive correlation). The white asterisk represents correlations that are statistically significant (*$p < 0.05$).

were found, of which down-regulated were 535 predicted genes, 112 genes encoding vomeronasal-1 and -2 receptors and 255 olfactory receptors. Despite the poor ability to classify the genes into bona fide functional classes by GO, KEGG and IPA, due to a high variability of the male transcriptomic data within the different treatment groups (see SI methods), upon manual annotation we could strikingly observe that genes that were common (155 in 45HFD and 24 in 60HFD) to the lists of DGEs in Cb and in ACX, except 4, were regulated in the opposite directions (Supplementary Table 5 in Supplementary Data 2 and Fig. 6d).

These included genes related to metabolism, inflammation and synapses (Fig. 6b, e). In addition, the mitochondrial proton carrier Ucp2 involved in oxidative stress, was significantly up-regulated (9.62-fold) in Cb of 45HFD along with ATP synthase mitochondrial F1 complex assembly factor 2 (Atpaf2), while it was down-regulated (-4-fold) in ACX.

In the cerebellum of female mice on 60HFD of 1325 transcripts, significantly altered 81 were vomeronasal-1 and -2 receptors, 120 are olfactory receptors and 372 are predicted genes, HFD45 produced 828 DGEs in female Cb, of which 192 were predicted genes, 36 vomeronasal-1 and -2 receptors and 67 olfactory receptors, all of which are down-regulated. Gene ontology (GO) analysis of up-regulated genes in 60HFD revealed a strong enrichment in biological processes, such as *regulation of cation channel activity*, *synapse organization*, *modulation of chemical synaptic transmission* and *regulation of dendritic spine morphogenesis*. GO annotation of down-regulated genes did not reveal any significant enrichment of any particular terms. Also in this case, of the 181 common DGEs in 60HFD Cb and ACX, 159 were regulated in the opposite directions, whereas for 45HFD Cb and ACX, 52 out of 68 common genes (Supplementary Table 6 in Supplementary Data 2). Also in Cb, in all conditions analyzed transcripts coding for a palmitoyltransferase were up-regulated (in males on 60HFD Zdhhc3, 2.01-fold; in males on 45HFD Zdhhc4 & 18 3.74- and 2.45-fold; in females on 60HFD Zdhhc5 2.21-fold). Venn diagrams and GO-terms of DEGs common to males and females are shown in Supplementary Fig. 8). In ACX for 785 common DEGs several GO-terms related to neuronal functions were found confirming analyses performed separately on males and females (Fig. 6c, f).

IPA analysis on female transcripts identified altered pathways, the most significant relating to synaptic function *SNARE signaling pathway* and *TypeII diabetes mellitus signaling* (common to both 45HFD and 60HFD), along with *synaptogenesis signaling* and *GABA receptor signaling pathway* (in 60HFD). Interestingly, the $z$-score of most identified pathways was positive, in particular the SNARE signaling pathway (2.65 in 45HFD, 3.9 in 60HFD) indicating up-regulation and thus confirming the results obtained by GO analysis and upon manual inspection of the DEGs lists (Supplementary Fig. 7 and Supplementary Data 1). Altogether the transcriptomic analysis reveals similar deregulations in male and females, which are opposite in ACX and Cb.

**Western blot analysis reveals sex dependent effects of HFD on AMPA/NMDA receptors and AKT functions**. Given the

modifications in genes involved in nervous system plasticity, synaptic function and neurotransmission in both sexes, we further investigated changes in neurotransmission-relevant proteins. Results revealed a significant decrease of AMPA GluA2/3 ($p = 0.024$) and GluA1 ($p = 0.0037$) in males but not in females fed with 60HFD compared to STD (Fig. 7a, b and Supplementary Data 1). Similar results were observed also for NMDA-R subunits (NR1 $p = 0.0048$; NR2A $p = 0.0028$) for the 60HFD vs STD group comparison (Fig. 7c, d and Supplementary data 1). Post-synaptic density protein 95 (PSD95) levels did not change after HFD consumption neither in males nor in females. Furthermore, based on sex specific differences found in metabolic functions and pathways, we assessed total and phosphorylated AKT protein levels. We found a significant increase in both total AKT and pAKT/AKT levels in 60HFD female mice (total AKT: $p = 0.0178$; pAKT/AKT: $p = 0.048$) compared to STD group (Fig. 7e, f and Supplementary Data 1).

## Discussion

MetS has been shown to affect men and women differently, but this has been studied in particular with regard to the cardiovascular risks[19]. To our knowledge, a limited number of studies reported the effect of HFD on brain metabolic and gene expression profiles and none of them included the evaluation of sex after prolonged exposure to HFD.

**Systemic metabolic effects of HFDs**. Our results confirm that 45% and 60% HFD regimens induce obesity and MetS in a time- and sex-dependent manner[20,21] in adult mice. In particular, males were more vulnerable than females considering peripheral markers. They reached earlier than females significant body weight increase upon both diets. Both diets progressively modified cholesterol levels and liver enzymes with a more pronounced effect in males, reaching higher serum concentrations compared to females. Despite these differences, glucose intolerance at 31 weeks was similar, independently of sex and HFD diet. In a study focused on the effects of a high-fat high-sugar diet, Garcia-Serrano et al. showed that glucose intolerance was more severe in male than female mice after 24 weeks of diet but no differences in brain metabolites were detected, suggesting a major relevance of nutrient composition than calories in homeostatic effects of diets[22].

In agreement with Lizarbe et al.[23] in males, the effect of HFD on brain glucose uptake was more evident for 60HFD than for 45HFD.

**Brain glucose metabolism and correlates with systemic metabolic markers of HFDs**. In our study, an increase in [18F]-FDG uptake in olfactory bulbs was common to both sexes after 60HFD. Olfactory bulbs have a primary role in food control and exert a direct impact not only on weight gain but also on the regulation of peripheral metabolism[24]. Furthermore, in males we found also a significant alteration of glucose metabolism in the anterior cortex and cerebellum, suggesting that male brain is more susceptible to metabolic changes induced by HFD. Interestingly, in Cb, brain glucose uptake modification was opposite to those of

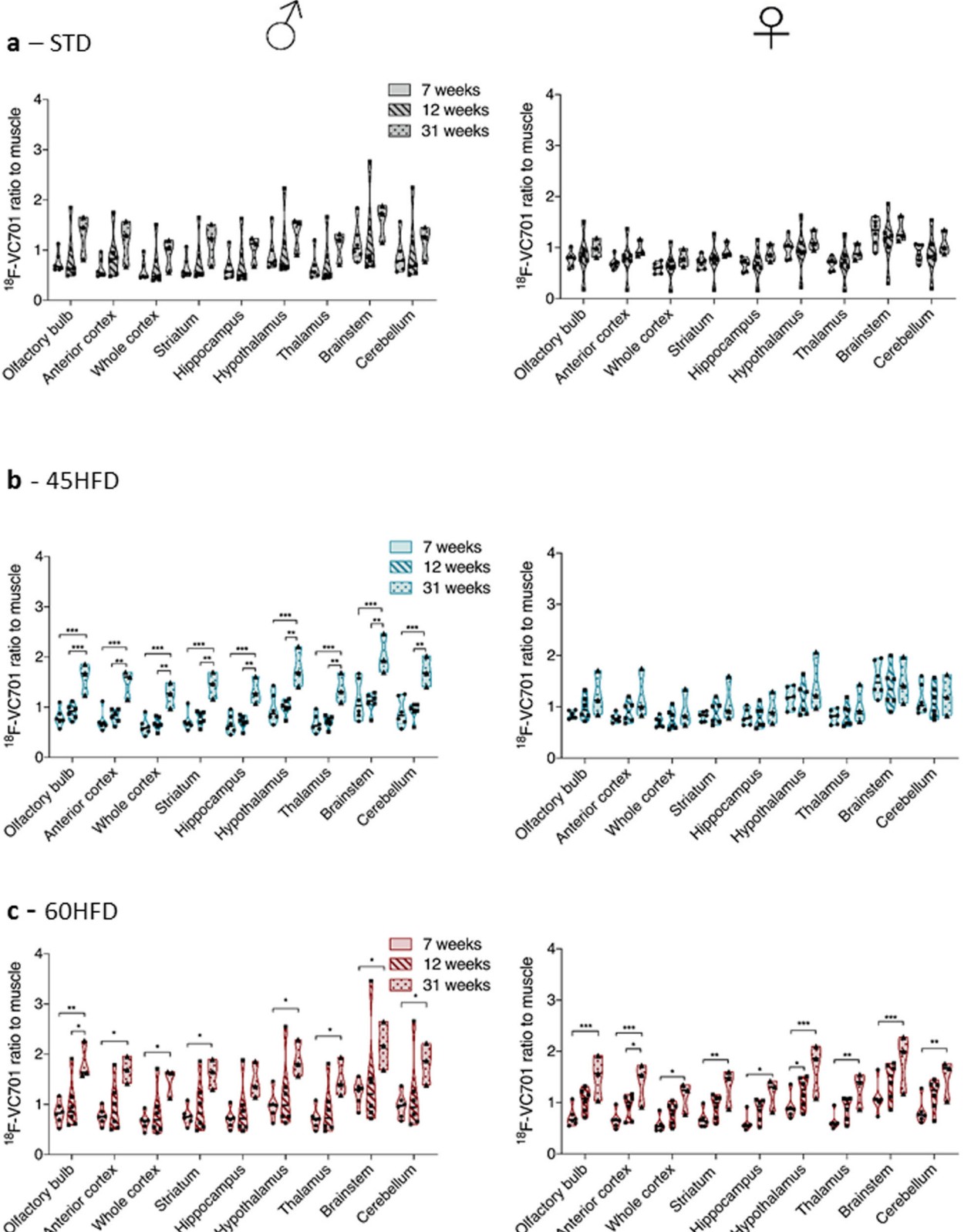

**Fig. 3 Longitudinal [$^{18}$F]-VC701 uptake ratio of different brain regions in male (left panels) and female (right panels) mice at 7, 12, and 31 weeks of diet. a** STD (control diet), **b** 45% HFD (45HFD), **c** 60% HFD (60HFD). Data are expressed as mean ± SD of ratio to muscle radiotracer uptake. All data were analyzed using two-way ANOVA in which within each row, each column mean was compared, followed by Tukey's post hoc test for multiple comparison (*$p \leq 0.05$; **$p \leq 0.01$; ***$p \leq 0.001$).

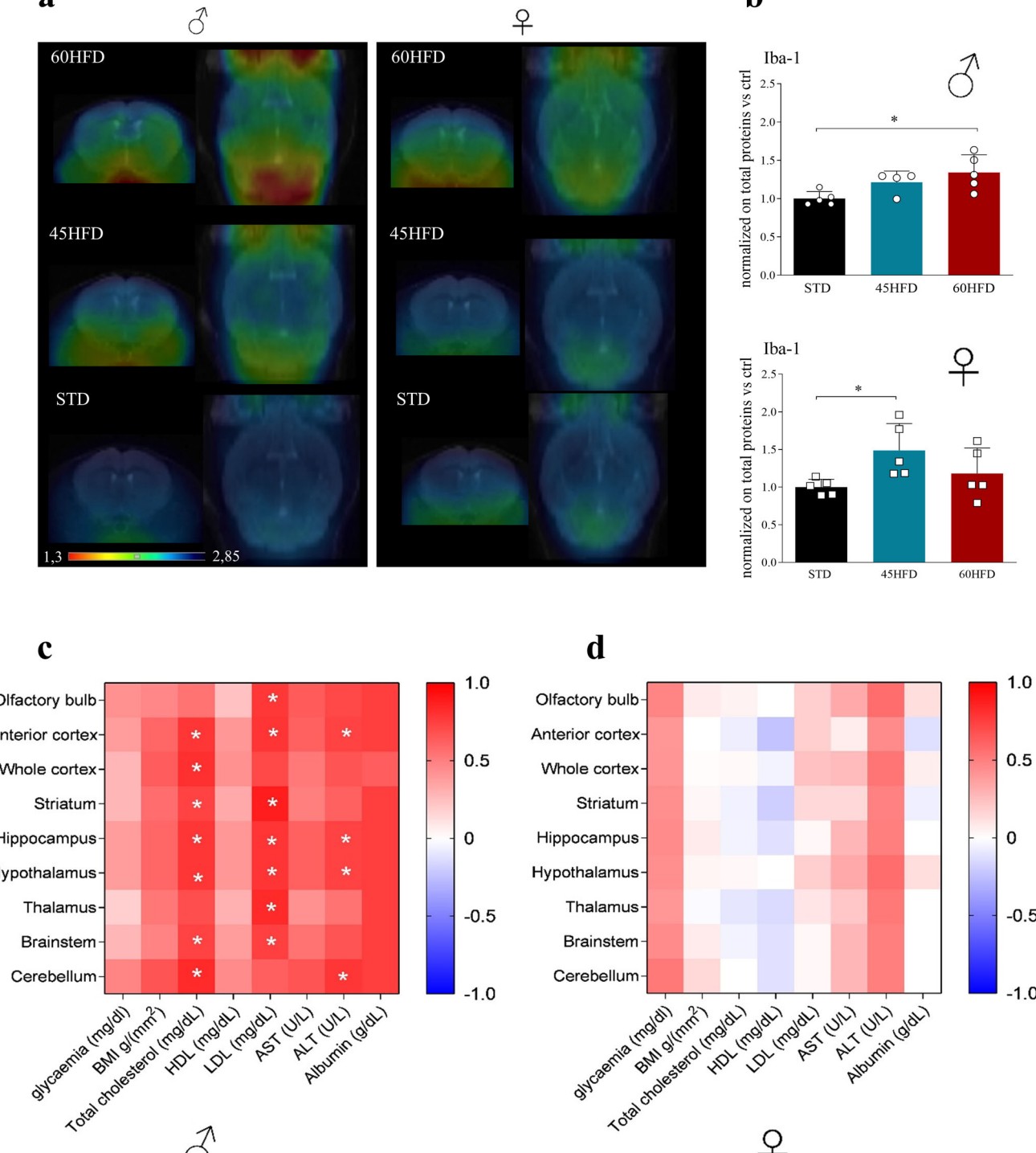

**Fig. 4 Brain neuroinflammation after HFD regimen. a** Representative panel of [$^{18}$F]-VC701 tracer brain uptake in male (left) and female (right) mice of all groups (STD, 45% HFD and 60% HFD) after 31 weeks. PET images are reported as brain to muscle ratio. **b** Western blot quantification of Iba-1 protein in male and female anterior cortex at 35 weeks of diet. Data, normalized on total proteins, are analyzed using Kruskal–Wallis test followed by Dunn's post hoc test and are expressed as mean ± S.E.M. vs STD (*$p \leq 0.05$). **c, d** Heat maps of Spearman's *r* correlation coefficients between [$^{18}$F]-VC701 brain uptake and basal glycemia, BMI, and serum level parameters at 31 weeks of diet in male (**c**) and female (**d**) mice. The pseudocolor bars represent the values ranging from −1 (blue, negative correlation) to +1 (red, positive correlation). The white asterisks represent correlations that are statistically significant (*$p \leq 0.05$).

other regions and was present also after 45HFD. Brain hypometabolism in selected regions is an established hallmark of Alzheimer's Disease (AD). However, several authors suggested that an increase in brain metabolism preceding the development of cognitive symptoms may act as a compensatory reaction to synapse dysfunction[25,26] an effect detected at the protein level in males ACX but not in females. Willette et al.[27] studied the relationship between regional modifications in glucose brain metabolism and insulin resistance in normal, mild cognitive impairment (MCI) subjects progressing or not to AD and patients

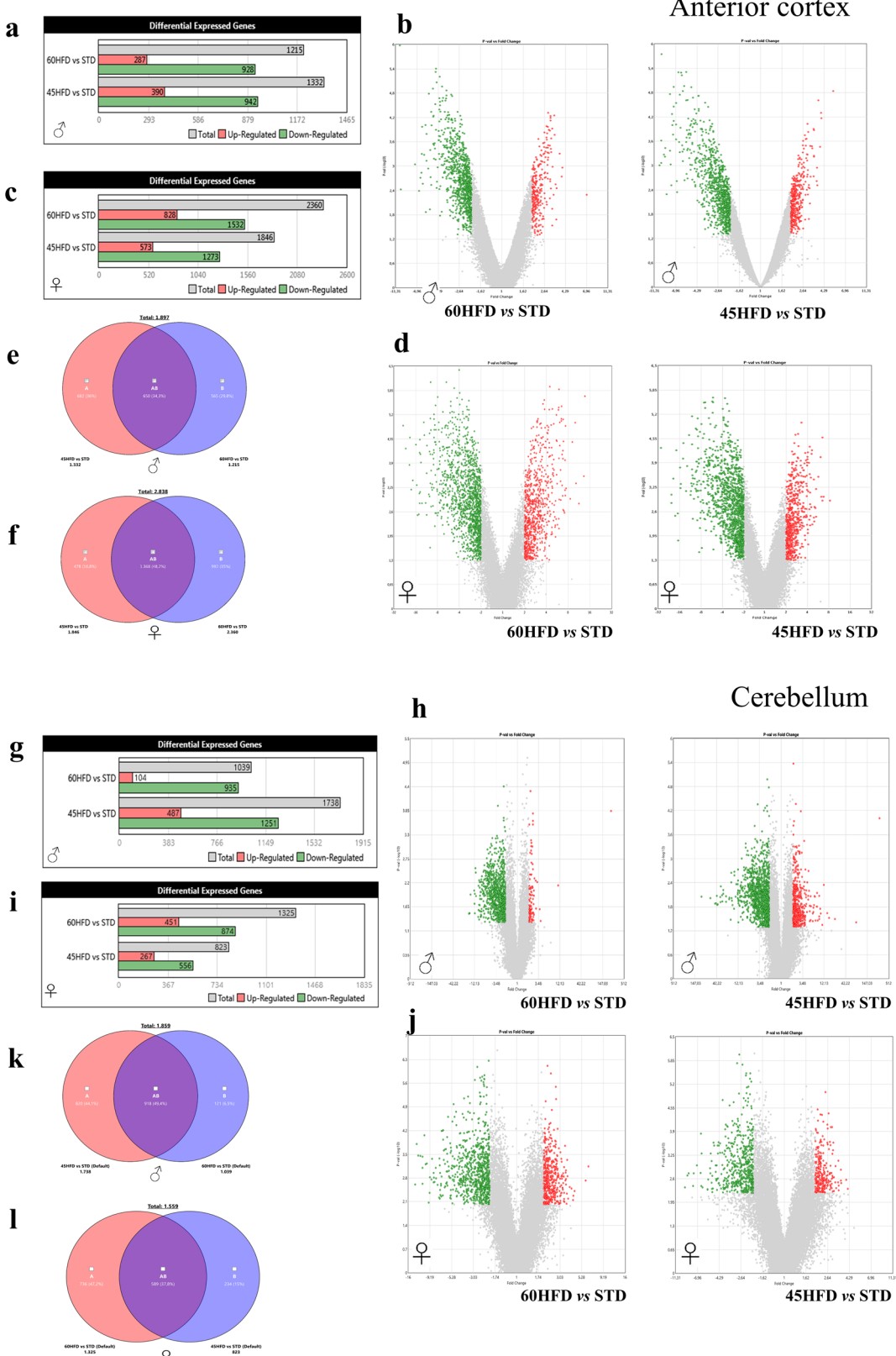

with AD diagnosis. He showed that in MCI subjects who progressed to AD, [18F]-FDG uptake was higher and associated with HOMA-IR. In another study, higher mean amyloid deposition in the precuneus was associated with hypermetabolism in MCI and hypometabolism in AD, suggesting that a transient increase in regional metabolism precedes the reduced metabolism characterizing AD[28]. Instead, the significant reduction in cerebellar metabolism observed in males after both 45 and 60HFD and the negative correlations with peripheral markers (cholesterol, HDL and BMI for males and ALT for females) observed in both sexes confirm vulnerability of the cerebellum to HFD already described in humans[29].

**Fig. 5 Anterior cortex and cerebellar transcriptomic profiles after HFD regimen. a–f** Histograms resuming anterior cortex DEGs in 60% HFD and 45% HFD mice including numbers of up- and down-regulated genes compared to STD control group in males (**a**) and females (**c**). Volcano plots representing the DEGs results of 60% HFD (**b** left) and 45% HFD (**b** right) compared to STD diet in males. Volcano plots representing the DEGs results of 60% HFD (**d** left) and 45% HFD (**d** right) compared to STD diet in females. Red dots indicate up-regulated genes while green dots indicate down-regulated genes that are statistically significant. Venn diagram of common and diet specific DEGs for the 60% and 45% HFD male (**e**) and female (**f**) mice. **g–l** Histograms resuming cerebellar differentially expressed genes (DEGs) in 60% HFD and 45% HFD mice including numbers of up- and down-regulated genes compared to STD control group in males (**g**) and females (**i**). Volcano plots representing the DEGs results of 60% HFD (**h** left) and 45% HFD (**h** right) compared to STD diet in males. Volcano plots representing the DEGs results of 60% HFD (**j** left) and 45% HFD (**j** right) compared to STD diet in females. Venn diagram of common and diet specific DEGs for the 60% and 45% HFD male (**k**) and female (**l**) mice. Red dots indicate up-regulated genes while green dots indicate down-regulated genes that are statistically significant. Data were analyzed using one-way ANOVA with the cut-off *p* value ≤0.05 and fold change ≥2.

Moreover, in males but not in females, whole brain [18F]-FDG uptake correlated inversely with glycemia, LDL and liver enzymes. Overall, these results indicate a sex-dependent brain metabolic response to prolonged HFD exposure.

**Imaging inflammation in brains of males and females**. Neuroinflammation measured with PET was not limited to the hypothalamus but was measurable in all analyzed brain areas of males exposed for 31 weeks to both diets and of females exposed to 60HFD. In addition, only in males, [18F]-VCF701 binding was correlated with total cholesterol, LDL and ALT levels. These data were partially confirmed by measurement of microglia Iba-1 marker showing a significant increase in males on 60HFD but not in females. This discrepancy cannot be explained by [18F]-VCF701 binding to astrocytic TSPO, since we did not detect any significant modification of GFAP levels in ACX but it is more probably related to the technique used: in vivo binding versus measurement of total protein tissue levels.

The partial difference between TSPO and IBA1 expression in female brain is not easy to explain. Iba-1 is widely used as a microglial activation marker and thus to proof in vivo results in post mortem analyses. A reduction of Iba-1, but not of the constitutive markers P2YR12 has been reported in obese subjects[30] as well as in a subset of microglial cells during neurodegeneration[31] suggesting that Iba-1 is differentially expressed in microglial phenotypes. Furthermore, according to some authors, in vivo TSPO signal predominantly reflects cell density and not activation phenotype[32]. Microglial cells exist in a broad spectrum of phenotypes that are often disease-state specific[33]. Due to the complexity of microglia markers expression a complete understanding of the neuro-inflammatory response associated to HFD may be obtained only in ad hoc studies based on the application of single cell analysis to microglia cells. An increase of TSPO-PET binding and microglial density was also shown in rats after a prolonged exposure to high glucose diet[34]. A diffuse microglial activation in response to prolonged HFD was already reported by some but not all authors[35]. Recently, Lizarbe et al.[23] evaluating the same HFDs as ours for 6 months in male mice reported a significant increase in Iba-1 levels only in the hypothalamus and limited to 60% HFD.

**Transcriptional effects of HFDs on anterior cortex and cerebellum**. Based on our PET results, showing opposite metabolic alterations following HFD we focused on the transcriptional profiles of two different brain regions: the anterior cortex (ACX) and the cerebellum (Cb), both involved in food regulation and behavior. To our knowledge, albeit few studies investigated the effect of a fatty acid rich diet in mice brains at molecular levels, none was focused on the anterior cortex and cerebellum of male and female mice.

The prefrontal cortex participates in different cognitive associative functions including the regulation of feeding.

Dysregulation of prefrontal cortex subregions reduces the activity of executive functions linked to caloric intake, increasing anxiety and appetite and thus favoring unhealthy eating behavior[36,37]. The Cb is also part of brain circuits controlling homeostatic and hedonic feeding through direct and indirect association pathways with hypothalamus, limbic system, and neocortical association areas[38,39]. Structural modifications of the cerebellum go frequently along with increase in BMI and gray matter volume of cerebellum, inversely correlating with abdominal obesity and related inflammatory processes[38]. In line with our findings, in a large study on obese subjects, gray matter density of different brain regions, including bilateral frontal lobes and cerebellum, was associated with BMI in males but not in females[40]. Despite the milder peripheral metabolic response of females, which was demonstrated also in the skeletal muscle of the same animal model after 14 weeks of HFD[41], ACX transcriptomic analysis revealed in both sexes, a prominent HFD-induced down-regulation of gene transcripts associated with nervous system development, axonal guidance, synaptic transmission, regulation of protein misfolding and BBB integrity, confirming previous evidence[42,43]. The down-regulation of excitatory synapse organizer Cerebellin1[40] observed only in males fed with 60HFD, may help explain the significant reduction of excitatory synapse receptor proteins observed in males. Our transcriptome analysis results agree with previous studies showing that obesogenic diets in males decreased GABA levels, dendritic spine density and changed microglial morphology in brain regions involved in cognition including prefrontal cortex[44,45]. Furthermore, in agreement with our results, 16 weeks feeding with 60HFD reduced the expression of genes related to neurogenesis, synapse, and calcium signaling in the cortex of male mice[46]. However, our results add an additional layer of complexity by showing that in Cb the opposite occurs both at the metabolic and transcriptional level, with neuronal and synaptic genes, oppositely to ACX, being up-regulated upon HFD. In both males and females, deregulated genes common to ACX and Cb were altered in opposite directions: these included olfactory receptor transcripts, genes involved in synaptic activity, inflammation and metabolism. In the Cb of HFD mice synaptic components are up-regulated, both inhibitory and excitatory synapses in females and mostly inhibitory in males indicating a possible excitatory-inhibitory imbalance typically observed in neuropsychiatric disorders[47].

In addition, in the ACX, we found transcript modifications previously observed in neurodegenerative disorders. For instance, reduction of Lrp in neurodegenerative disorders favors plaque formation and is linked to Parkinson's and Lewy body diseases[48]. Moreover, HFD increased $A_{2A}R$ in males while reducing CX3CL1/Fractalkine and its receptor CX3R1 in females, suggesting dysregulation of endothelial and microglial functions[49,50]. $A_{2A}R$ overexpression impaired short-term object recognition memory and working memory[51].

Up-regulation of Nlrp5 in females and 9b in males ACX, which are expressed by cerebral endothelial cells and pericytes and are

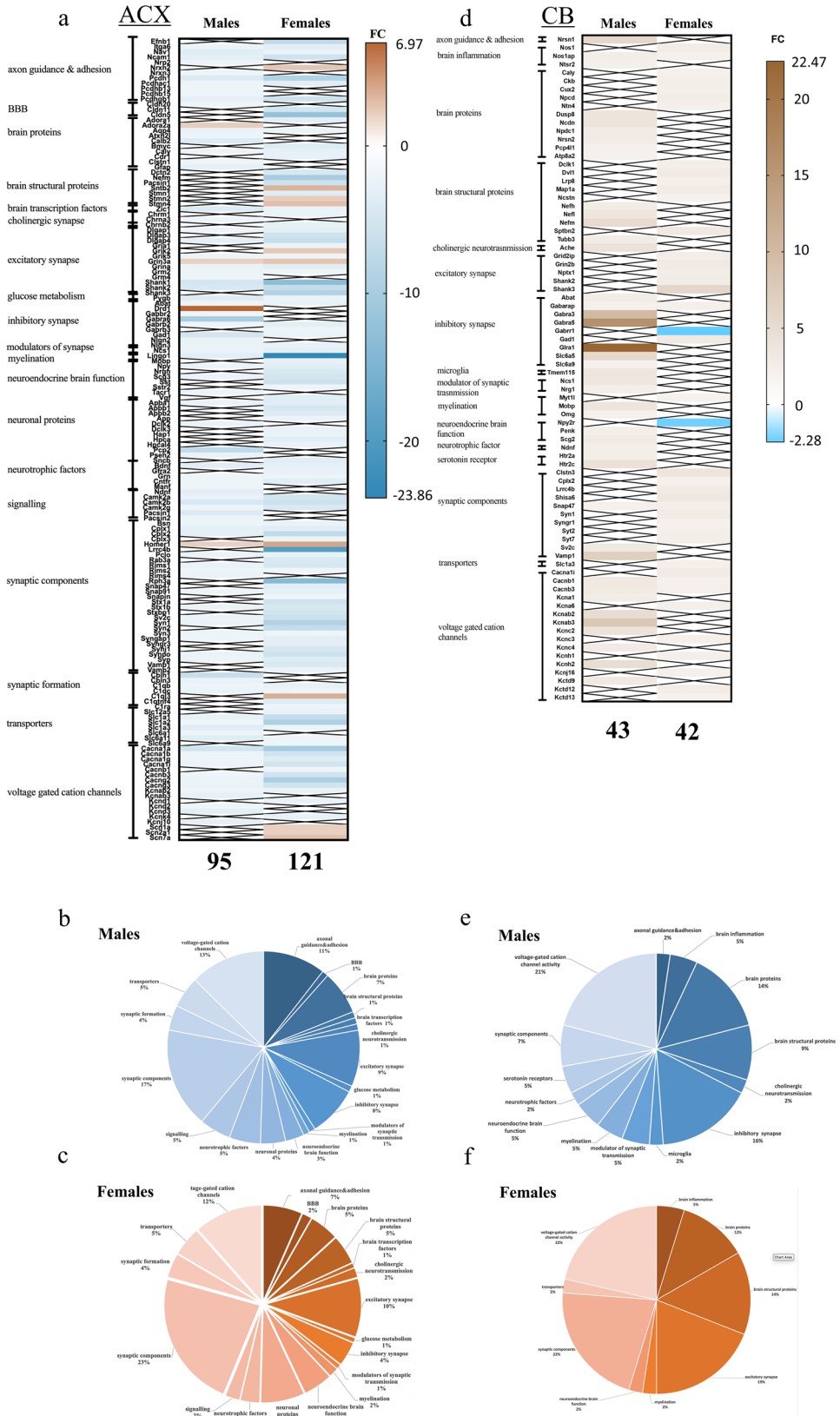

up-regulated following inflammatory stimuli[52,53], confirms an ongoing inflammatory process in both sexes, possibly mediated by different types of stimuli. Overall prolonged exposure to HFD modified brain transcript expression in both sexes showing a higher variability in males compared to females. Modification of brain transcripts, particularly those related to neuronal functions,

was region specific and oppositely regulated in ACX and Cb in both sexes.

**Implications for synaptic functions.** Evaluation of glutamatergic AMPA, GLUA2/3 and GLUA1 and NMDA, NR1 and NR2A receptor subunits in the ACX, showed significant differences in

**Fig. 6 Differentially expressed neuronal genes in anterior cortex (ACX) and cerebellum (Cb) of females and males. a** Heatmap representing the different functional classes of genes found within neuronal DGEs in ACX of males and females on HFD60 vs STD. **b, c** Pie diagrams showing the representation of different gene categories among the neuronal DGEs in males (blue) and in females (brown). **d** Heatmap representing the different classes of neuronal DGEs in Cb of males on HFD45 and females on HFD60 vs STD. **e, f** Pie diagrams showing the representation of the different gene categories among the neuronal DGEs in males (blue) and in females (brown). The pseudocolor bars shows the fold change: up-regulated (brown, max value 6.97 in **a**, max value 22.47 in **d**), down-regulated (light blue, max value −23.86 in **a**; max value −2.28 in **d**). The X in the Heatmap indicates the absence of the gene in the list.

males but not females fed on HFD, while p-AKT/AKT levels were increased in females but not in males. In astrocytes, AKT and p-AKT regulate glycogen synthesis and mitochondrial activity, with increased p-AKT levels indicating an adaptive astrocyte response to brain insults[54]. On the other hand, modulation of AKT pathways also regulates dendritic density of glutamatergic synapses and this may explain the lack of modification in glutamatergic receptors in female anterior cortex[55]. Moreover, females exhibited modifications in AMPK pathway and in the levels of Pygm and Pdks, enzymes that control glycogen utilization and mitochondrial OXPHOS activity. AMPK regulates neuronal metabolic plasticity by increasing glycolysis during synaptic activation[56] and a reduction of AMPK related transcripts is in line with the lack of [18F]-FDG uptake increase in the ACX of female mice. Only in females, target genes of HIF1a are altered, such as the mitochondrial fatty acid transporter Slc25A5 and glut1 transporter, Slc2A1, were down-regulated, while genes involved in glycolysis, such as Pdk1 and Ldhb were up-regulated. Pdk1 is a hypoxia inducible gene, which is critical for the attenuation of mitochondria ROS production, diminishing the flow of pyruvate to the TCA cycle, but maintaining ATP levels and adapting to hypoxia[57]. Thus, in females, an apparent reduction in glucose production by glycogenolysis and an impaired handling of pyruvate occurs, in parallel with increase in lactate production with no evidence of elevations in lactate utilization. On the contrary, in males, the increase of the MCT transporter Slc16a5, suggests a shift in the metabolism towards lactate.

Interestingly, up-regulation of AKT and down-regulation of AMPK, attenuate the effects of IR exerted by HFD in the hypothalamus[58,59] thus potentially representing a neuron-protective strategy, which might be specific to females.

**Functional considerations**. The relationship between anterior cortex and cerebellar regions in feeding control is not fully understood. Interestingly, the simultaneous stimulation of prefrontal cortex and inhibition of the cerebellum by transcranial direct current stimulation (tDCS) influences appetite in obese individuals. In particular, prefronto-cerebellar neuromodulation increased hunger and desire to eat and reduced working memory performance, without modifying basic motor performances[60]. This result suggests a role of cerebellum in higher cognitive functions and an opposite connectomic regulation between the two regions, which is in line with our findings based on opposite metabolic (male) and transcriptomic (males and females) regulations in anterior cortex and cerebellum. Our data suggest an important relationship between these two regions not only in brain functions related to food intake but also in the response to impaired glucose tolerance, a result of unhealthy food consumption.

Females appear to be more protected systemically from metabolic diseases than males when exposed to fat diets for a long time. However, the brain transcriptomic analysis, revealed that females are similarly impacted, with pathways related to synaptic functions modified similarly in both sexes, albeit synaptic receptor proteins were altered only in males.

**Age and metabolic alterations**. The complex interaction between obesity and risk of developing cognitive decline is likely modulated by age. The metabolic effect of diet-induced obesity and the relative differences between sexes was shown to depend on the age of onset particularly regarding glucose tolerance[61]. In addition, aging or neurochemical modifications related to AD exacerbated in males the effects of obesity on brain plasticity and inflammation[62]. Regarding the brain, the effects of age is more complex. Different epidemiological findings suggest a protective effect of high BMI for dementia after the age of 65 years[63]. However, a long history of obesity is also considered a risk factor[64]. For this reason, our brain molecular results are in line with studies that suggest that overweight or obesity increase the risk for brain dysfunction. Moreover, the many transcript alterations of female brain in response to prolonged HFD may explain why overweight AD females show a higher vulnerability and a reduced neuronal reserve following brain metabolism dysfunctions, recently described in a large PET metabolic study on male and female AD patients[65]. The key to understand the sex-specific response to MetS might lie in the different gene expression alterations between males and females. Starting from these pieces of evidence, further research is needed to investigate more deeply the mechanisms involved in genes deregulation and its sex-dependent impact on cognition and behavior.

Altogether, our results indicate a sex-dependent susceptibility to metabolic alterations following a prolonged HFD feeding, with males being more affected than females in peripheral organs despite similar glucose intolerance. At the transcript level in ACX, HFD down-regulated genes coding for synaptic proteins and functions in both sexes, but in the Cb synaptic genes were up-regulated. How the molecular modifications observed in the brain of females could explain the higher susceptibility to Alzheimer disease warrant some considerations.

## Methods

**Ethical statement**. All animal studies were carried out in accordance with the European Union Directive for Protection of Vertebrates used for Experimental and Other Scientific Ends (2010/63/EU) and institutional guidelines for the care and use of experimental animals.

**Animals and diets**. Five-week-old male and female *C57Bl/6J* mice (Charles River, Italy) were housed in the San Raffaele Institute animal facility, maintained in a 12/12-h light/dark cycle with access ad libitum to food and water. One week after acclimation, mice were randomly assigned to one of the following diets: Standard diet (STD) (D12450B, Research Diets Inc); 45% HFD (45HFD) (D12451, Research Diets Inc.) and 60% HFD (60HFD) (D12492, Research Diets Inc.), containing respectively 10%, 45% and 60% of calories derived from fat (Supplementary Table 1). A total of 45 male and 45 female mice were included. Longitudinal PET studies were performed at 7, 12, and 31 weeks of diet using [18F]-FDG and [18F]-VC701 as radiopharmaceuticals. Sacrifice time was 14 weeks for one group of animals ($n = 30$) and 33 weeks ($n = 30$) for the second one; sample size for each experiment is specified in Supplementary Information Table 2.

A separate group of 30 mice were sacrificed at the end of 14 weeks and dedicated to liver immunohistochemistry and serum analyses. The same group was part of a study already published on HFD induced muscle damage[41].

Animals body weight was monitored weekly. BMI was calculated as body weight g (body length$^2$ mm$^2$)$^{-1}$ measured from the tip of the muzzle to the attachment of the tail. Caloric and energy intake (kcal day$^{-1}$) were also calculated, weighing the food daily (gram day$^{-1}$ consumed). A schematic diagram of each experiment is reported in Fig. 8 and number of animals per task in Supplementary

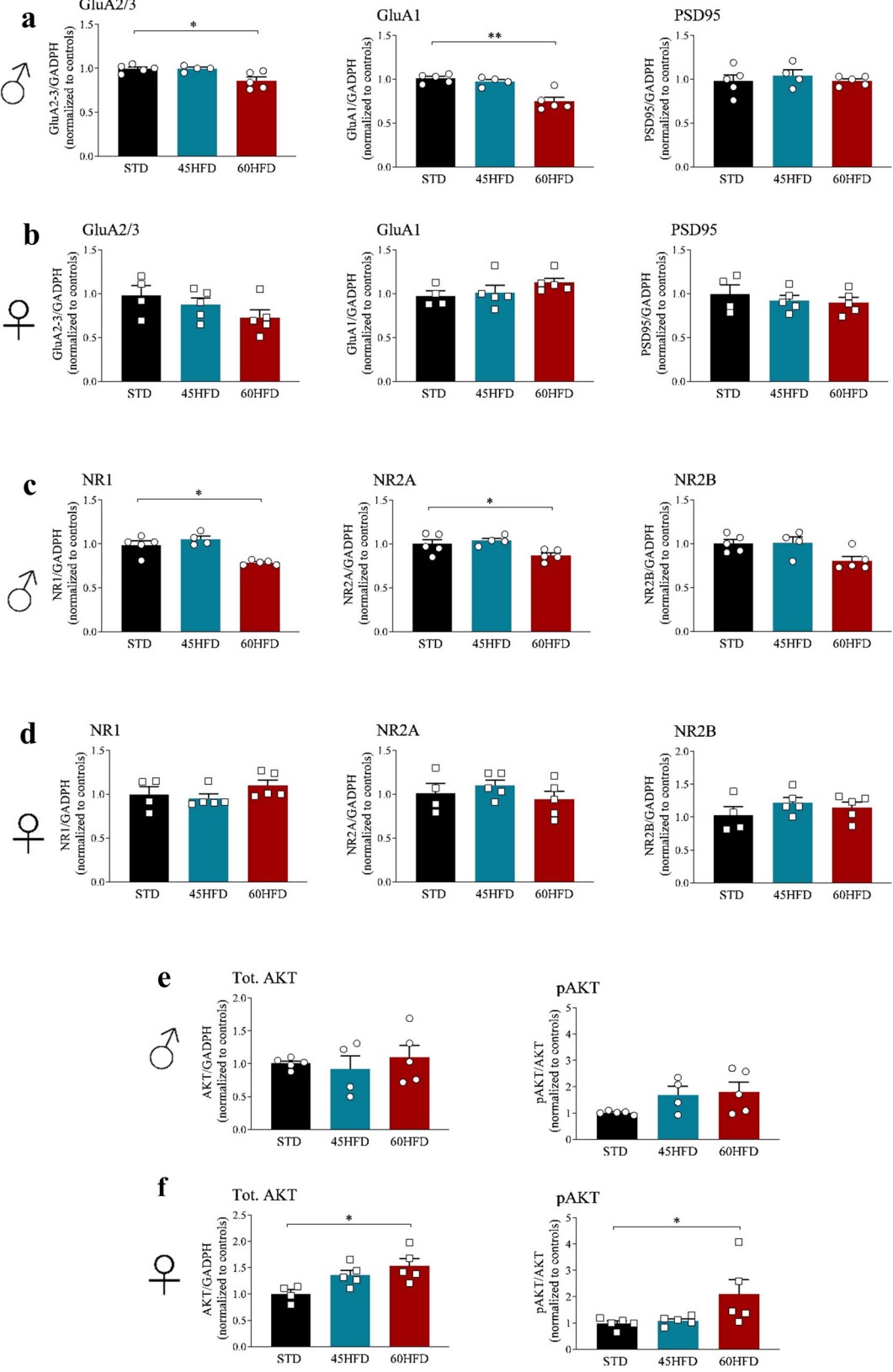

**Fig. 7 Post mortem analysis of anterior cortex. a** Western blot analysis of AMPA receptor subunits: GLUA2/3, GLUA1 and PSD95 protein in anterior cortex of male mice and female mice **b** at 33 weeks of diet. **c** Western blot analysis of NMDA receptors subunits NR1, NR2A, NR2B, in the anterior cortex of male and female mice **d** at 33 weeks of diet. **e** Total AKT and phosphorylated AKT in male and female mice **f** at 33 weeks of diet. Data are analyzed using Kruskal–Wallis test followed by Dunn's post hoc test and are expressed as mean ± S.E.M. of normalization on total proteins vs control (*$p ≤ 0.05$).

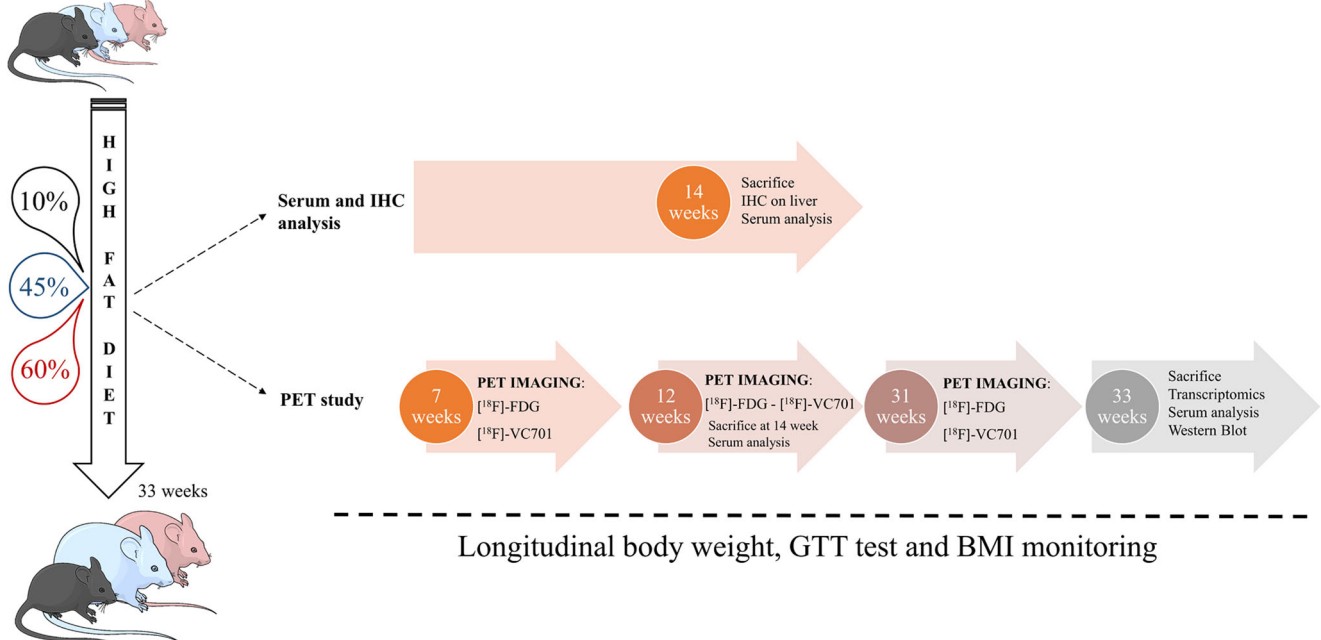

**Fig. 8 Schematic representation of the experimental study design showing on the left the three experimental groups fed with three different diets varying in their lipidic contents (10% STD, 45% HFD45, 60% HFD60).** On the right, a diagram represents the studies conducted on the three experimental groups and the time points. Two main sets of experiments were conducted, as indicated by the arrows. In the first experiment, animals on the three diet regimens were sacrificed at 14 weeks and analyzed for biochemical parameters (serum evaluation) and liver immunohistochemistry (IHC); in the second experiment, animals underwent longitudinal PET imaging studies and were sacrificed at 33 weeks of the diet regimen for ex vivo investigation including transcriptomic, biochemical serum evaluation and western blotting. All the animals used in the study were monitored longitudinally for body weight, glucose tolerance and body mass index (BMI).

Information Table 2. On the basis of the presence of multiple outcomes, minimum sample size was calculated using the resource equation[66].

**Metabolic evaluation**. Mice were monitored for IP-GTT at 31 weeks. Basal glycemia was measured during fasting conditions at the end of PET-FDG studies, collecting a blood drop from the tail, after application of local anesthetic, using a glucometer (StatStrip Xpress®2, Nova Biomedical, MA, USA) at 0 (before glucose injection), 15, 30, 45, 60, and 120 min after i.p. glucose injection (2 g kg$^{-1}$ of bw, dissolved in saline).

**Hemato-chemical serum analyses**. Serum analyses were performed at 14 and 33 weeks of diet. Immediately after sacrifice by vertebral dislocation after a light isoflurane anesthesia (2% in air), blood was collected by heart puncture into Eppendorf tubes, clotted for 30 min and serum collected by centrifugation (1300 × $g$, 10 min). Samples were stored at −80 °C until use. Total cholesterol (mg dL$^{-1}$), HDL, (mg dL$^{-1}$), LDL (mg dL$^{-1}$), aspartate aminotransferase (AST; UI L$^{-1}$), alanine aminotransferase (ALT; UI L$^{-1}$) and albumin (g dL$^{-1}$) were determined by the LAM Service (Laboratory of Murine Analyses facility) at our institution. Internal quality control was done for every test type. Serum analyses were measured by spectrophotometry (ILab ARIES, Werfen Instrumentation Laboratory S.p.A., Milan, IT) according to the manufacturer's protocol. Hemolyzed serum samples were excluded.

**Liver immunohistochemistry**. The right anterior liver lobes of animals after 14 weeks of diets were fixed in formalin, paraffin-embedded and three/four sections per animal were stained with hematoxylin and eosin (H&E) to visualize the steatosis in the liver parenchyma.

**Brain PET imaging**. PET was performed on the YAP(S)-PET II small animal tomograph (ISE S.r.l., Pisa, Italy). Glucose metabolism was measured with [18F]-FDG (2-[18F]fluoro-2-deoxy-D-glucose), whereas brain inflammation with the Translocator Protein (TSPO) targeting agent, [18F]-VC701 as previously shown[67,68]. [18F]-FDG is a glucose analog widely used to measure regional brain metabolism. [18F]-FDG enters in the brain thanks to glucose transporters and is than phosphorylated in 6 C position by hexokinases and trapped into the brain cells according to metabolic demand[69,70]. [18F]-VC701 is a structural derivative of [11C]PK11195 with higher affinity for TSPO, developed by our group and already applied in preclinical settings to measure neuroinflammation. During resting conditions, brain TSPO levels are low. However, thanks to its overexpression on activated microglia, infiltrating macrophage, and reactive astrocyte. For these reasons TSPO has been widely used as a target for the in vivo imaging of neuroinflammation in different brain disorders using PET[71].

Animals were injected with 4.31 ± 0.05 MBq of [18F]-FDG and or with 4.80 ± 0.22 MBq [18F]-VC701 and acquired at 30 or 120 min. respectively, after radiopharmaceutical injection in a tail vein. After positioning, the brain was centered in the PET gantry and acquired for 30 min. under 2% isoflurane in air. This anesthesia protocol was used to maintain mice asleep for all the duration of PET scan. PET [18F]-VC701 scan was performed in the afternoon with animals fasting from the morning, whereas [18F]-FDG was performed in the morning after an overnight fasting period. Minimum intervals between PET [18F]-FDG and [18F]-VC701 was 48 h starting the first PET exam during the 7th, 12th, and 31th week after the beginning of the diet. Acquisition order was randomly assigned between groups. Data were corrected for the physical decay of fluorine 18 ($t_{1/2}$: 109.8 min) and calibrated to transform count per pixel in MBq per mL. Image analysis was performed using PMOD 2.7 software (Zurich, Switzerland). Images were manually co-registered to an MRI template and radioactivity concentration extrapolated using a Region Of Interest (ROI) brain template automatically positioned on co-registered images [30]. For this reason, blinded analysis was not applied. Radioactivity concentration uptake was expressed as SUV (standardized uptake value) and obtained by dividing radioactivity concentration expressed in MBq mL$^{-1}$ of tissue to the injected dose and by multiplying it for mice body weight. Regional [18F]-FDG data were normalized to mean whole brain uptake values as performed in clinical studies. [18F]-VC701 uptake was normalized to muscle, since no reference brain region could be selected.

**Statistics and reproducibility**. Male and female mice were directly compared for peripheral metabolic response. Brain effects were evaluated separately in males and females. Minimum sample size for PET studies was calculated using a separate set of brain [18F]-FDG-PET data, using G*Power software and considering a power of 0.8, alpha of 0.05 and an estimated effect size calculated considering a diet induced relevant modification equal to 3 Standard Deviation of the mean value of uptake in the anterior cortex. Given these parameters, total sample size for three conditions was equal to 9. Statistical evaluation (GraphPad Software Inc., CA, USA) was performed using one-way analysis of variance (ANOVA) when comparing the three groups of dietary regimens at the same time point, two-way ANOVA for comparison of two variables including the diet and sex and/or time point while three-way ANOVA was used to asses sex, diet and time point differences. GTT and body weight was tested with one-way ANOVA with repeated measures. All tests were followed by Tukey's or Dunnett's post hoc test for multiple comparisons ($p \leq 0.05$). Correlations were performed using Spearman's $r$ analysis. Data were

summarized into a heat map and expressed as *r* coefficient. The asterisk has been inserted for *r* values >|0.7| and *p* < 0.05. Reproducibility of western blot analyses was assessed by performing at least two technical replicates of all experiments and using the technical replicates in the statistical analysis.

**Post mortem brain analyses**. Molecular evaluations were performed by transcriptomic (anterior cortex and cerebellum) and western blot analyses of mice anterior cortex tissues collected after the last PET study. Anterior cortices of each hemisphere were dissected, snap-frozen in liquid nitrogen and stored at −80 °C. One hemisphere was dedicated to transcriptomic, the other for protein analyses. For transcriptome analysis before RNA extraction, tissues were stabilized in RNA Later™ (Thermo Fisher) on ice, at −20 °C ON, and then disrupted and homogenized in RLT buffer (350 μL) using TissueLyser (Qiagen). Western blot analysis was performed on anterior cortices collected after 33 weeks of HFD or control diets.

**Transcriptomic analysis**. Total RNA was extracted following the Qiagen RNeasy Plus Micro kit (Qiagen Italia) procedure for the anterior cortex and by the Directzol™ RNA MiniPrep (Zymo Research) for the cerebellum. The RNA concentration and purity were measured by spectrophotometer (Nanodrop) and 260/280 and 260/230 ratios measured. RNA integrity was assessed by Agilent Bioanalyzer calculating the RNA Integrity Number (RIN), where good quality had a RIN ≥ 6 and the presence of clear 28s and 18s rRNA peaks. An aliquot (10 ng) of RNA was used for the preparation of targets for Clariom™ S Mouse arrays, according to the IVT Pico Reagent kit (Thermo Fisher). The Clariom™ S Mouse arrays (Thermo Fisher Scientific; Massachusetts, USA) contain >22,100 genes, >150,300 transcripts, >221,900 total probes with several probes targeting genes >221,300. The staining, washing and scanning of the arrays were conducted using a Fluidics 450 station, Command Console Software and GeneChip® Scanner 3000 7G, generating.CEL files for each array. The images were analyzed by Thermo Fisher GeneChip Command Console (AGCC) and analyzed with the Thermo Fisher GeneChip Expression Console. The quality control of the scanned data was first estimated by confirming the order of the signal intensities of the Poly-A and Hybridization controls using Expression Console Software (Thermo Fisher Scientific, USA). Raw expression values were imported as Thermo Fisher.CEL files into TAC 4.0 software (Thermo Fisher Scientific, USA). PCA on cortical and cerebellar sample of male and female was performed as well as Venn diagram analyses.

The anterior cortex transcriptome analysis in male mice was performed by analyzing 14 samples (5 STD, 4 45HFD, and 5 60HFD). PCA was performed and concordant biological replicates corresponding to 2 biological samples for the group STD and 45HFD and 4 samples for the group 60HFD were selected for further analysis. For the experiments on female animals a total of 15 samples (5 STD, 5 45HFD and 5 60HFD) were uploaded and normalized. PCA was again used to select the most concordant biological replicates within each experimental group and 9 samples were selected 3 chips for each STD, 45HFD and 60HFD groups for further analysis.

A Robust Multichip Analysis (RMA) quantification method[72] was used as a probe set summarization algorithm for log transformation with base 2 (log2) and Quantile normalization method was chosen to evaluate the preliminary data quality in the Preprocessing module. The mean signal intensities of all genes were obtained. After normalization, the differentially expressed genes satisfying the conditions of the fold change cut-off 2 and a one-way analysis of variance (ANOVA) with adjusted *p* value <0.05 from all the genes probed in the array, were selected as differentially expressed genes (DEGs).

The same methodological approach was applied to cerebellum samples. For females, PCA showed as concordant 12 samples (5 60HFD, 4 45HFD, and 3 STD). The probes were tested for differential expression using One-Way Analysis of variance (ANOVA), and adjusted *p* values of <0.05 and FC of ≥absolute 2.0 were applied. Exploratory Grouping Analysis (EGA) indicated that the male cerebellum samples form 2 distinct clusters (C1 and C2) with 3 samples in C2 belonging to each of the condition group studied (1×STD, 1×60HFD, and 1×45HFD) indicating that these three samples are much more different within their treatment group. In general, male samples showed high variability compared with what observed with female samples. Of note, two STD samples (the two mice with the highest weight gain) displayed a variance similar to those of the 60HFD and 45HFD group and 2 STD samples were well separated from the other groups. High variability was also observed within the 45HFD samples (2 mice were shown to have a very different variance between the other 3 mice present in the group). Based on these observations, we grouped the samples accordingly to the EGA and PCA results. Those biological replicates that were more concordant to each other were selected for further analysis. Using a high-stringency (FDR-adjusted) statistical analysis approach to account for multiple comparisons testing, no genes were found to demonstrate differential expression between the different HFD conditions. When the data were re-analyzed using a similar statistical approach, without FDR-adjustment, a number of genes were identified as differentially expressed following HFD in the brain cerebellum but only when the 2 STD mice samples, which were well separated were used. The final group of mice were the following: −60HFD (3 mice); −45HFD (3 mice). Applied thresholds were 2-fold change and *p* value <0.05. The lists of DEGs from both males and females were analyzed for ontology

enrichment using Database for Annotation, Visualization and Integrated Discovery (DAVID; http://david.abcc.ncifcrf.gov/) and the Enrichr web site (https://maayanlab.cloud/Enrichr/). Similarly, common DEGs found among male and female mice fed on 45HFD vs STD and 60HFD vs STD in the anterior cortex and cerebellum were analyzed for ontology enrichment and also common DEGs present in Cb and ACX of males and females.

Differentially expressed genes from females and males between 60HFD and STD were analyzed by the Ingenuity Pathway Analysis (IPA) software (QIAGEN Inc., https://www.qiagenbioinformatics.com/products/ingenuity-pathway-analysis in order to obtain information about altered pathways[73]. Most significantly pathways were those with a −log(*p* value)>1.3 and *z*-scores >|1.5| in at least one experimental group. Highly significantly altered pathways both in females and males, for which a z-score could not be calculated were also included in the list.

**SDS-PAGE and western blotting**. After thawing, the tissues (male *n* = 5 60HFD; 4 45HFD; 5 STD: female *n* = 5 60HFD; 5 45HFD; 4 STD) were homogenized twice for 3 min each, using the TissueLyser II (QIAGEN, Italy) in a lysis buffer containing 290 mM sucrose, Tris 0.5 M pH 6.8 and 3% SDS, 10 μg μL$^{-1}$ of each protease inhibitors (leupeptin, chymostatin, pepstatin A, and aprotinin), phosphatase inhibitors: sodium orthovandate 1 mM, Na F 10 mM, and Tetrasodium Pyrophosphate 10 mM. Lysates were centrifuged at 1530 × *g* for 8 min at 10 °C to remove insolubilized debris. The supernatants were recovered and diluted to a concentration of ~1 mg mL$^{-1}$ with assay buffer. Protein concentrations were measured using the BCA Protein Assay Kit (Pierce™, Thermo Fisher). Antibodies (Abs) used for anti-GluA1 and GluA2/3 AMPA receptor subunits were developed and characterized in house, as described[74]. The other NMDA receptor subunits were detected using and anti-NR1 (clone 54.1, BD Pharmingen, San Diego, USA), anti-NMDAR2 (NR2A, clone A3-2D10, Life technologies, Waltham, MA, USA) and anti-GluN2B (NR2B, clone N59/20; Antibodies Incorporated, Davis, CA, USA). We also used anti-PSD95 Ab (clone K28/43, Antibodies Incorporated, Davis, CA, USA), anti-human AKT1 mouse IgG1 (Invitrogen, Thermo Fisher Scientific, Monza, Italy), anti-human pSer473-AKT1 rabbit IgG (Thermo Fisher Scientific, Monza, Italy), Iba-1 Ab (WAKO_016.20001, Fujifilm, Osaka, Japan) and anti-GAPDH Ab (sc 25778, Santa Cruz Biotechnology, Dallas, Texas, USA). SDS-PAGE and blotting were carried out by standard procedures. In brief, 5–10 μg of proteins obtained from lysed tissues were separated on SDS-polyacrylamide gel electrophoresis using 7.5% acrylamide and electrophoresis-mediated transfer to nitrocellulose membranes with 0.45 μm pores (Schleicher and Schuell ll, Dassel, Germany). The blots were blocked overnight in 4% non-fat milk in Tris-buffered saline, washed in a buffer containing 4% non-fat milk and 0.3% Tween20 in Tris-buffered saline, and incubated for 2 h with the primary antibody at the following concentrations: GluA1, GluA2/3: 1–2.5 μg mL$^{-1}$, GluN2A 1:500, GluN1 1:1500, GluN2B 1:600, PSD95 1:4000, AKT and pAKT 1:1000, Iba-1 1:1000, GAPDH 1:4000. Primary Abs were then incubated for 1 h with the appropriate secondary antibody (anti-rabbit Ly-Cor IRDye800RD) and after several washes, membranes were dried overnight in the dark, at room temperature (RT). The infrared (IR) signal was measured using an Odyssey CLx - Infrared Imaging System. The Western blot bands of AMPA, NMDA receptor subunit, PSD and AKT/pAKT signal intensity were quantified using iStudio software normalizing to the GAPDH content and then to the optical density of STD control value (1) as previously described[1]. Iba-1 blots were performed by separating protein lysates on TGX Stain-Free™ precast acrylamide gels (gradients 4–15%) for SDS-PAGE. Proteins were transferred to nitrocellulose membranes (Nitrocellulose Transfer Kit Cat. #170–4270) using the Trans Blot Turbo system (Bio-Rad). Blots were incubated with 5% BSA blocking buffer followed by overnight incubation @4 °C with primary antibody (WAKO 016–2000) and secondary antibodies conjugated with HRP. Bioluminescent signals were acquired by Gel Doc Imaging system (Bio-Rad) and analyzed by Image Lab 6.1 software (Bio-Rad) by normalizing to total protein content provided by the TGX stain-free technology. Statistical analysis was obtained by applying Kruskal–Wallis test followed by Dunn's post hoc test with *p* value ≤0.05, as a cut-off. Data were expressed as mean values ± S.E.M. of 3–4 separate experiments performed in duplicate for each.

**Reporting summary**. Further information on research design is available in the Nature Portfolio Reporting Summary linked to this article.

## Data availability
The datasets generated during the current study supporting the findings of this study are available from the corresponding authors upon reasonable request. Source data used to generate figures are available as Supplementary Data 1. The raw data of transcriptomic analysis have been up-loaded onto GEO platform with the accession number GSE216442 (https://www.ncbi.nlm.nih.gov/geo/query/acc.cgi?acc=GSE216442).

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

## Acknowledgements

This work was supported by "AMANDA": Abnormal metabolic states, cellular stressors and neurodegenerative processes"—from Regione Lombardia (CUP_B42F16000440005), "NeON", "Platform for the identification of targets of pharmacological relevance for the treatment of pathologies of the nervous system and oncology with a high need for treatment" from Regione Lombardia-POR-FESR 2014–2020," ID 239047, CUP E47F17000000009, by the Italian Ministry of University and Research (MIUR)—Department of Excellence project PREMIA (PREcision MedIcine Approach: bringing biomarker research to clinic) and thanks to Euro-BioImaging (www.eurobioimaging.eu) for providing access to imaging technologies and services via the MultiModal Molecular Imaging Italian Node (Italy).

## Author contributions

R.M.M. and M.L.M. conceived the experiments, which were carried out by V.M. and S. Penati with the help of S.B., E.T., S.M., and R.C. C.G., S. Pucci, and S. Penati did the western blot analyses. R.M.M., M.L.M., and M.M. conceived the big project, which was written for obtaining the grants necessary for funding the experiments. M.L.M. and R.M.M. wrote the animal protocol, which was submitted by M.M. to the Italian ministry of health. V.M., S. Penati, and S.B. did the statistical analyses and the figures. M.F. and A.P. did the transcriptomic analysis and the statistical analysis concerning transcriptomic. R.M.M. and M.L.M. "digested" the transcriptomic analyses to generate figures and to write results and discussion. The manuscript was written by R.M.M., M.L.M., M.M., V.M., and S. Penati.

## Competing interests

The authors declare no competing interests.
