## [Peer Review File · Communications Biology]

Reviewers' comments:

Reviewer #1 (Remarks to the Author):

this manuscript reports sex differences upon long term HFD exposure. The manuscript collects a number of analyses, including PET measurements of glucose metabolism. Methodology is state of the art, although results are biased by limited sample size. The major issue I find in this study is that although the authors claim to study gender effects, but those are not statistically tested in the present study design. While some diet-induced phenotype alterations are similar in both sexes, others are different. In some cases, differences are masked by the above-referred small sample size. Therefore it is difficult to know whether any gender differences (not tested) are biologically relevant.

It is great that transcriptomics was followed up by a few individual analyses. However again, data suffers from small sample size.

Altogether this study concludes that males are more affected than females, but transcriptomics is mostly altered in females. Any reasoning of what can be done out of this needs further clarification.

In some datasets, 45% diet had stronger effects than 60% diet. This does not seem to be a general pattern observed in previous studies, and some controversy might be around. The authors should survey the literature in depth to understand all findings from this study. The study also misses key published studies. A study by Lizarbe et al. *Frontiers Neuroscience* 2019, testing the same diets but only in males should be compared to the present findings. Another study comparing sexes in diet-induced obesity should also be discussed here (Garcia-Serrano et al., *Aging and Disease*, in press)

Reviewer #2 (Remarks to the Author):

This study provides new information on sex differences in the effects of a chronic high fat diet on the brain. Administration of the high fat diet long term is a strength in that it has high clinical relevance. Including 2 HFD groups also allows for more nuanced investigation of the HFD effects. The PET imaging was also a major strength. However, there were a few weaknesses that need to be addressed.

Major comments:

1. Throughout the paper the authors use the term gender to refer to rodents. Gender is a social construct. Sex is the biological term. The term sex should be used for rodent studies. This needs to be corrected in the abstract and throughout the manuscript.
2. The entire paper is lacking in reporting sex differences (via 3 way ANOVA). This is a major issue that needs to be addressed.
3. Even though many of the results shown are in the anterior cortex, what is already known about the anterior cortex and HFD is missing from the discussion, as is the rationale for why this region is important.
4. The "take home message" for each figure could have been clearer.
5. Overall, the discussion lacks interpretation of what their results might mean with regards to sex differences in HFD's influence on the development of neurodegenerative diseases, even though various diseases were mentioned
6. Sex differences in the metabolic effects of a high fat diet are highly dependent on the age of onset. Here a juvenile onset model is used, which is known to cause more severe metabolic impairment in males than females (while later onset does not, DOI: 10.1038/s41366-018-0023-3). The implications for this in influencing the results should be discussed.
7. Figure 2A, B, and E were analyzed via 2-way ANOVA but should be analyzed with repeated

measures ANOVA

Minor comments:

1. The radioligands used should be described better when introduced in the methods.
2. Results section corresponding to Figure 2 says "HFD diet induces insulin resistance and progressively leads to metabolic modifications resembling metabolic syndrome". "HFD diet" is redundant, it should say "HFD". There is no direct measure of insulin, there is only the indirect measure of glucose tolerance
3. Area under the curve analyses for GTT results would be helpful.
4. Line 166: possible typo—do they mean 1215 or 121 genes?
5. Figure 7A, the color scale for the heat map would be easier to read if it were a 1- or 2-color gradient, not a rainbow
6. Line 270: the link that the authors are making between the clinical study of MCI and their HFD effects in the brain is sudden.
7. Line 275-277: the switch to discussing the cerebellum is also sudden "...suggests a higher vulnerability of the cerebellum to HFD". Is this in comparison to the anterior cortex, or to the brain as a whole?
8. Figure 3E and F and Figure 5C and D use red/green color matrices, which makes it difficult for a red/green colorblind person to distinguish the direction of weaker correlations
9. Figure 7A, females appear on the left and males appear on the right, which is inconsistent with the other graphs in the paper
10. The text in figures 2-3 of supplement is too small to read on the figures.

Reviewer #3 (Remarks to the Author):

This is an interesting manuscript focused on sex differences in the brain and periphery to prolonged high fat diet challenge. The manuscript has multiple strengths, including the multi-modal (imaging, histology, molecular, metabolic) evaluation. Results from the study add to the body of evidence on the need of taking into account sex when peripheral and, importantly, central effects of metabolic challenges, such as HFD, are considered translationally.

The manuscript does have several deficiencies, some quite important.

1. The title of the manuscript is misleading as it suggest that the cortex was the some focus whereas majority of the data was generated across multi0le brain regions.
2. Similarly, the abstract of the manuscript is not a true reflection of the study's experimental design and results; it is a highly selective subsample.
3. Fig. 8 refers to ex vivo analyses, but presents WB data from post mortem samples; the term ex vivo implies that live tissue was extracted, maintained viable, and then subjected to some kind of treatment prior to analyses.
4. Insulin sensitive/resistance was not evaluated and this is a substantial limitation.
5. In females, the correlation between TSPO PET and IBA-1 is not apparent; the authors have not provided reasonable explanation for this, nor have they followed this result with a more detailed microglia vs. Macrophage analysis.
6. Given the different directionality of the HFD effect on brain glucose uptake between the anterior cortex and the cerebellum, molecular signature of the transcript changes in the cerebellum and how they compare to the ones in the cortex, which were more pronounced in the females, is desirable.
7. Correlation analyses (Fig3, EF, and Fig 5, CD; these analyses are too simple and do not capitalize on the fact that glucose uptake and microglia/macrophages were imaged in the brains of the same mice. Is there any correlation between glucose uptake and neuroinflammation in the context of any of the correlations with the peripheral biomarkers?
8. At the end of the study, the mice were at the cusp of what is considered middle age: was estrous

cycle monitored, was it affected by diet, and was glucose uptake affected by age (or Sex by age) in the control mice?

Point-to-point reply to reviewer's comments concerning the manuscript "**Brain sex-dependent alterations after prolonged high fat diet exposure**" (previously entitled "Sex-dependent alterations in anterior cortex after prolonged high fat diet exposure") by Valentina Murtaj, Silvia Penati, Sara Belloli, Maria Foti, Angela Coliva, Angela Papagna, Cecilia Gotti, Elisa Toninelli, Remy Chiaffarelli, Stefano Mantero, Susanna Pucci, Michela Matteoli, Maria Luisa Malosio and Rosa Maria Moresco. We thank reviewers for their suggestions to improve our manuscript. We have thoroughly addressed all issues and we hope to have done a satisfactory work. In our opinion the manuscript is now much clearer and the claim and the novelty are much more understandable.

Concerning reviewer's comments here are the point-to-point replies.

Reviewers' comments:

Reviewer #1 (Remarks to the Author):

1. this manuscript reports sex differences upon long term HFD exposure. The manuscript collects a number of analyses, including PET measurements of glucose metabolism. Methodology is state of the art, although results are biased by limited sample size. The major issue I find in this study is that although the authors claim to study gender effects, but those are not statistically tested in the present study design. While some diet-induced phenotype alterations are similar in both sexes, others are different. In some cases, differences are masked by the above-referred small sample size. Therefore it is difficult to know whether any gender differences (not tested) are biologically relevant.

As suggested by the reviewer, for peripheral effects, we repeated ANOVA analysis including also sex and time as variables (Results Fig. 2, Supplementary Fig.1 and 2 rows 120-138 and Methods row 532-543). Brain effects were evaluated separately in males and females as indicated in the statistical analysis paragraph of the revised version of the manuscript (Methods rows 532- 537).

The study was designed in general to compare males and females for large effects in the periphery. Two way and repeated way ANOVA analysis were implemented in the revised version to include sex and diet or sex, time and diet in the comparisons. In light of the exploratory nature of the study and the complexity of PET studies, males and females were considered separately.

We have included in the Statistics and Reproducibility section (rows 533- 537) the following statement: "Minimum sample size for PET studies was calculated using a separate set of brain [¹⁸F]-FDG-PET data, using G*Power software and considering a power of 0.8, alpha of 0.05 and an

estimated effect size calculated considering a diet induced relevant modification equal to 3 Standard Deviation of the mean value of uptake in the anterior cortex. Given these parameters, total sample size for three conditions was equal to 9”.

Revised Figure 2

Figure 2

New Supplementary Figure 1

Supplementary Figure 1: BMI of male and female mice after 7 and 12 weeks of high fat diet consumption

(a) BMI in males and females at 7 weeks of diet. (b) BMI in males and females at 12 weeks of diet. Data were analyzed using Two-Way ANOVA including sex and diet as variable and were expressed as mean \pm SD. * $p \leq 0.05$; *** $p \leq 0.001$

Revised Supplementary Figure 2

Supplementary Figure 2: Biochemical parameters of females and males fed with high fat diets for 14 and 33 weeks

45HFD (left panel) and 60HFD (right panel) vs STD male and female at 14 (14w) and 33 weeks (33w); **(a)** High Dense Lipoprotein (HDL); **(b)** Low Dense Lipoproteins (LDL); **(c)** Aspartate transaminase enzyme (AST); **(d)** Alanine transaminase enzyme (ALT); **(e)** Albumin (Alb). Data were analyzed using Three-Way ANOVA followed by Tukey's post hoc test for multiple comparison using sex and diet as covariate and expressed as mean \pm SD.

\$ significance between HFD males and HFD females at 14 weeks; # significance between HFD males and HFD females at 33 weeks.

2. It is great that transcriptomics was followed up by a few individual analyses. However again, data suffers from small sample size.

The number of samples included in the transcriptomic analysis are explained thoroughly in the new version of the Methods, rows 575- 588 for anterior cortex “The anterior cortex transcriptome analysis...(DEGs).”, and rows 589- 617: “The same methodological approach...were also included in the list.”). Unfortunately, animals, mostly males in the cerebellar transcriptomic analysis, showed a greater variability than females which reduced the number of DEGs obtained, possibly underscoring the differences between diets and controls. For details see rows 595-608 (“In general, male samples....The final group of mice were the following :...(3 mice).”) in the Methods section.

3. Altogether this study concludes that males are more affected than females, but transcriptomics is mostly altered in females. Any reasoning of what can be done out of this needs further clarification. Upon revision and inclusion of transcriptomic data of the cerebellum, as suggested by reviewer#3, we have come to the conclusion that males are more affected than females in the periphery and also in synaptic proteins of the anterior cortex. Despite this, surprisingly females were equally affected to males at the transcriptomic level, both in anterior cortex and in the cerebellum. In the latter tissue however the DEGs were regulated in the opposite way as in the anterior cortex. This has been inserted in the revised version of the manuscript in the results section (rows 180-272 and new figures 6, 7, Suppl. figure 7, Suppl. Table 5, 6 and has been discussed in the paragraph starting at row 343 (Transcriptional effects of HFDs on anterior cortex).

New Figure 6

Figure 6

New Figure 7

Figure 7

4. In some datasets, 45% diet had stronger effects than 60% diet. This does not seem to be a general pattern observed in previous studies, and some controversy might be around. The authors should survey the literature in depth to understand all findings from this study. The study also misses key published studies. A study by Lizarbe et al. *Frontiers Neuroscience* 2019, testing the same diets but only in males should be compared to the present findings. Another study comparing sexes in diet-induced obesity should also be discussed here (Garcia-Serrano et al., *Aging and Disease*, in press).

The aim of our study was to investigate the overall peripheral and CNS metabolic dysfunction related to high fat diet consumption. We took in consideration two diets with different fat composition as it is was not clear from the literature, as you pointed out, which one represents the best model to study MetS. We observed a similar effect of 45% and 60% HFD on regional brain metabolism in the cerebellum of male mice with a general higher brain variability in males than females that reflected peripheral responses. This was confirmed also in the transcriptomic analysis. For the other variables measured, the effects was in general higher for 60% HFD. From the results we have obtained, we can consider the 60HFD as a solid model to investigate both peripheral and CNS alterations, both in male and in female mice. We are aware of a certain variability in the response of mice to HFDs, exemplified by the literature that uses essentially 3 types of diets, the high fat diet with 45% containing 35% Carbohydrates and 20% proteins, the high fat diet with 60% of energy from fats with 20% Carbohydrates and 20% proteins and the western diet with 40% of energy from lipids 43% from carbohydrates and 17% from proteins. Carbohydrates, in particular sucrose in the diet may play an important role in stimulating insulin release and its action on peripheral tissues, so for example the 45HFD contains 17% of calories from sucrose compared to the 60HFD, which contains only 6.7%, whereas the western diet contains 29% of calories from sucrose. This could play an important role in the overall effect of the diet on metabolism and its long-term effects. A comprehensive discussion of this aspect would deserve a review article, we therefore opted not to include too much of this topic in the discussion.

With all this reasoning in mind, we have made the effort, beside comparing males and females responses, to compare also diets, which was in some cases a bit disappointing as you pointed out, and would deserve a more appropriate investigation. We are aware that interindividual variability might play an enormous role, also single animals might effectively eat different amounts of the diet on a daily basis. A more detailed study with a more controlled diet administration method should be performed in order to clarify, which of the two diets works best for the MetS model. In the revised version we have included the suggested references and discussed the key studies suggested by reviewer, as well as others. We have added also the study of Garcia-Serrano et al. that was not yet

published when our manuscript was submitted (rows 298-303). Conclusions have been modified to more clearly describe study results. We have extensively rewritten the discussion and conclusions including more references. New pieces of data have been added according to reviewer's suggestion.

Reviewer #2 (Remarks to the Author):

This study provides new information on sex differences in the effects of a chronic high fat diet on the brain. Administration of the high fat diet long term is a strength in that it has high clinical relevance.

Including 2 HFD groups also allows for more nuanced investigation of the HFD effects. PET imaging was also a major strength. However, there were a few weaknesses that need to be addressed.

Major comments:

1. Throughout the paper the authors use the term gender to refer to rodents. Gender is a social construct. Sex is the biological term. The term sex should be used for rodent studies. This needs to be corrected in the abstract and throughout the manuscript.

Reviewer is right, according to the reviewer's indication, gender has been substituted with sex throughout the manuscript.

2. The entire paper is lacking in reporting sex differences (via 3 way ANOVA). This is a major issue that needs to be addressed.

As suggested also by reviewer#1, for peripheral effects, we repeated ANOVA analysis including also sex and time as variable : Fig.2 c and g were analyzed using Two-Way ANOVA including sex and diet as variable. Data in Fig. 2 d and e were analyzed using Three-Way ANOVA, in which diet, sex and time point were considered. For the analysis presented in Supplementary Fig.2 a Three-Way ANOVA with diet and sex were also applied. Brain effects were evaluated separately in males and females as indicated in the statistical analysis paragraph of the revised version of the manuscript (rows 531-545).

3. Even though many of the results shown are in the anterior cortex, what is already known about the anterior cortex and HFD is missing from the discussion, as is the rationale for why this region is important.

The rationale for selecting anterior cortex and the reference to previous data have been included in the introduction (rows 109-112: “Anterior cortex was selected for molecular analysis, because it is involved in cognitive function related to food preferences, eating disorders and anxiety. Cerebellum was added to transcriptomic analysis on the basis of the PET metabolic results and of volumetric modification reported from imaging studies on obesity.”) and in the discussion section (rows 352-358 “The Cb is also part of brain circuits... was associated with BMI in males but not in females”) of the revised manuscript as well as reference to published data concerning cerebellum (rows 354-358; rows 413-420: “The relationship between anterior cortex and cerebellar regions... regulations in anterior cortex and cerebellum”) that has been subjected to transcriptomic analysis according to reviewer#3 suggestion.

4. The “take home message” for each figure could have been clearer.

We included take home messages for each figure in the corresponding result section:

Fig. 2 rows 137-138; Fig. 3 rows 159-162 ; Fig. 4 and 5 rows 177-180; Fig. 6 and 7 rows 278-279; for Fig. 8 it has been included in the title of the results paragraph, see rows 281-282.

5. Overall, the discussion lacks interpretation of what their results might mean with regards to sex differences in HFD’s influence on the development of neurodegenerative diseases, even though various diseases were mentioned.

We have thoroughly revised the discussion along with reviewer’s suggestions. In the revised version of the manuscript, we hope to have appropriately addressed and discussed the potential associations between our results and an increased risk of dementia (rows 434-448: “However, a long history of obesity is also considered a risk factor... higher susceptibility to Alzheimer disease warrant some considerations.”).

6. Sex differences in the metabolic effects of a high fat diet are highly dependent on the age of onset.

Here a juvenile onset model is used, which is known to cause more severe metabolic impairment in males than females (while later onset does not, DOI: 10.1038/s41366-018-0023-3). The implications for this in influencing the results should be discussed.

Age dependent effects as well as the study indicated by the reviewer have been reported and discussed in the revised version (rows 427-448: “Age and metabolic alterations...”).

7. Figure 2A, B, and E were analyzed via 2-way ANOVA but should be analyzed with repeated measures ANOVA

In the revised manuscript Fig. 2, data in panels a, b and e, which now is f (GTT) were analyzed with repeated measure analysis, whereas AUC analysis (panel g) was analysed by using Two-Way ANOVA including sex and diets as variables. (rows 532-543: “Statistics and Reproducibility... All tests were followed by Tukey's or Dunnett's post hoc test for multiple comparisons ($p \leq 0.05$).”).

Minor comments:

1. The radioligands used should be described better when introduced in the methods.

Radioligands properties have been described in the methods section of the revised version of the manuscript (rows 504-511).

2. Results section corresponding to Figure 2 says “HFD diet induces insulin resistance and progressively leads to metabolic modifications resembling metabolic syndrome”. “HFD diet” is redundant, it should say “HFD”. There is no direct measure of insulin, there is only the indirect measure of glucose tolerance.

The sentence has been modified accordingly and insulin resistance has been substituted with Glucose Tolerance. In a parallel independent study, which is underway, using the same animal model but only males, we have performed also the Insulin tolerance test and it turns out that mice under the same dietary regimens of 45 and 60HFD develop insulin resistance after 5 weeks of HFD.

3. Area under the curve analyses for GTT results would be helpful.

AUC have been calculated and included in Fig. 2g of the revised version of the manuscript.

4. Line 166: possible typo—do they mean 1215 or 121 genes?

1215 was correct

5. Figure 7A, the color scale for the heat map would be easier to read if it were a 1- or 2-color gradient, not a rainbow.

We have modified the color scheme of Fig 7 according to reviewer's suggestion and have used a 2-color gradient representation (light blue and brown). We hope to have chosen appropriate colors for highlighting the different patterns of gene regulation in anterior cortex and cerebellum.

6. Line 270: the link that the authors are making between the clinical study of MCI and their HFD effects in the brain is sudden.

This part has been modified and we hope that is now more exhaustive (rows 316-322).

7. Line 275-277: the switch to discussing the cerebellum is also sudden "...suggests a higher vulnerability of the cerebellum to HFD". Is this in comparison to the anterior cortex, or to the brain as a whole?

We intended the comparison with the whole brain. The role of cerebellum in feeding and the main finding deriving from imaging studies have been included and discussed in the revised version (rows 322-326: "Instead, the significant reduction in cerebellar metabolism observed...humans.", 369- 376: "However, our results add an additional layer of complexity... observed in neuropsychiatric disorders.").

8. Figure 3E and F and Figure 5C and D use red/green color matrices, which makes it difficult for a red/green colorblind person to distinguish the direction of weaker correlations

This figure has been modified accordingly.

9. Figure 7A, females appear on the left and males appear on the right, which is inconsistent with the other graphs in the paper.

Figure 7 has been substantially revised including also the aspect pointed out by the reviewer. See above in the point-to-point letter.

10. The text in figures 2-3 of supplement is too small to read on the figures.

Figures have been generated by the Enrichr website and are difficult to modify, unless we split them and generate more figures with less panels.

Reviewer #3 (Remarks to the Author):

This is an interesting manuscript focused on sex differences in the brain and periphery to prolonged high fat diet challenge. The manuscript has multiple strengths, including the multi-modal (imaging, histology, molecular, metabolic) evaluation. Results from the study add to the body of evidence on the need of taking into account sex when peripheral and, importantly, central effects of metabolic challenges, such as HFD, are considered translationally.

The manuscript does have several deficiencies, some quite important.

1. The title of the manuscript is misleading as it suggest that the cortex was the some focus whereas majority of the data was generated across multiple brain regions.

The title has been modified according to reviewer's suggestion.

2. Similarly, the abstract of the manuscript is not a true reflection of the study's experimental design and results; it is a highly selective subsample.

The abstract has been rewritten completely and we hope that it now satisfactorily reflects the claim of the manuscript, including also new transcriptomic data on cerebellum.

3. Fig. 8 refers to ex vivo analyses, but presents WB data from post mortem samples; the term ex vivo implies that live tissue was extracted, maintained viable, and then subjected to some kind of treatment prior to analyses.

The title of figure 8, as well as the Methods section (row 549) referring to post mortem samples have been modified according to reviewer's suggestion.

4. Insulin sensitive/resistance was not evaluated and this is a substantial limitation.

We limited our selection to GTT as surrogate marker of metabolic impairment because we didn't want to stress animals too much. In fact, animals were submitted to several imaging sessions at different time points and adding also Insulin tolerance measurement appeared a little too much. As commented in response to reviewer#2 minor point 2, we know from a parallel study that male animals on the same high fat diets develop insulin resistance 5 weeks after HFD exposure.

5. In females, the correlation between TSPO PET and IBA-1 is not apparent; the authors have not provided reasonable explanation for this, nor have they followed this result with a more detailed microglia vs. Macrophage analysis.

Differences between TSPO and Iba-1 have been discussed (rows 335-342: "This discrepancy cannot be explained by [¹⁸F]-VCF701 binding to astrocytic TSPO,... Iba-1 levels only in the hypothalamus and limited to 60% HFD.") and the analysis of the astrocytic marker GFAP have been included in supplementary results of the revised version.

6. Given the different directionality of the HFD effect on brain glucose uptake between the anterior cortex and the cerebellum, molecular signature of the transcript changes in the cerebellum and how they compare to the ones in the cortex, which were more pronounced in the females, is desirable.

We have performed the transcriptomic analysis of the cerebellar tissue as suggested and the results are included in the revised version of the manuscript (rows 236-272: “Cerebellum (Cb) transcript analysis... Altogether the transcriptomic analysis reveals similar deregulations in male and females, which are opposite in ACX and Cb.”). We find an interesting parallelism between the opposite FDG glucose uptake and the differential genes. In fact, we see genes common to anterior cortex and cerebellum, which are regulated in the opposite way (Suppl Table 5 and 6, results rows 236-272, discussion rows 416-422: “In particular, prefronto-cerebellar neuromodulation..., a result of unhealthy food consumption.”). We thank reviewer for the suggestion, which has opened a new interesting path to be further investigated in the future.

7. Correlation analyses (Fig3, EF, and Fig 5, CD; these analyses are too simple and do not capitalize on the fact that glucose uptake and microglia/macrophages were imaged in the brains of the same mice.

Is there any correlation between glucose uptake and neuroinflammation in the context of any of the correlations with the peripheral biomarkers?

No correlations were found between peripheral glucose uptake and brain inflammation. TSPO and FDG correlated with different peripheral markers.

8. At the end of the study, the mice were at the cusp of what is considered middle age: was estrous cycle monitored, was it affected by diet, and was glucose uptake affected by age (or Sex by age) in the control mice?

In the study, we did not monitor estrous. No time or sex dependent effect in FDG uptake was observed in control mice.

REVIEWERS' COMMENTS:

Reviewer #1 (Remarks to the Author):

All my comments have been addressed, and the manuscript has been modified accordingly. Despite the small sample sizes, the transcriptomics datasets would be highly valuable for many researchers, if freely available for additional analyses. That would also substantially increase the use (citations) of this article.

Reviewer #2 (Remarks to the Author):

All comments were appropriately addressed

Reviewer #3 (Remarks to the Author):

In the revision, the authors have addressed many of the issues raised in the reviews of the original manuscript. New data (cerebellum) and new statistical analyses were done. However, some of the data that are amenable to more sex effects analyses are not subjected to such. One example is Figure 7 where it is unclear what the commonalities/differences between the sexes in the ACX and the CB are; Venn diagrams with overlapping pathways and genes between the sexes and the two brain regions would provide clarity. Possible sex differences in TSPO cell type expression within the brain and the lack of congruency between IBA-1 and the PET data should be elaborated.